# Associations between compensable injury, perceived fault and pain and disability 1 year after injury: a registry-based Australian cohort study

Melita J Giummarra,[1,2,3] Katharine S Baker,[3,4] Liane Ioannou,[4] Stella M Gwini,[1] Stephen J Gibson,[3] Carolyn A Arnold,[3,5] Jennie Ponsford,[4,6] Peter Cameron[1]

For numbered affiliations see end of article.

**Correspondence to**
Dr Melita J Giummarra; melita.giummarra@monash.edu

## ABSTRACT

**Objectives** Compensable injury increases the likelihood of having persistent pain after injury. Three-quarters of patients report chronic pain after traumatic injury, which is disabling for about one-third of patients. It is important to understand why these patients report disabling pain, in order to develop targeted preventative interventions. This study examined the experience of pain and disability, and investigated their sequential interrelationships with, catastrophising, kinesiophobia and self-efficacy 1 year after compensable and non-compensable injury.

**Design** Observational registry-based cohort study.

**Setting** Metropolitan Trauma Service in Melbourne, Victoria, Australia.

**Participants** Participants were recruited from the Victorian State Trauma Registry and Victorian Orthopaedic Trauma Outcomes Registry. 732 patients were referred to the study, 82 could not be contacted or were ineligible, 217 declined and 433 participated (66.6% response rate).

**Outcome measures** The Brief Pain Inventory, Glasgow Outcome Scale, EuroQol Five Dimensions questionnaire, Pain Catastrophising Scale, Pain Self-Efficacy Questionnaire, Injustice Experience Questionnaire and the Tampa Scale of Kinesiophobia.

**Methods** Direct and indirect relationships (via psychological appraisals of pain/injury) between baseline characteristics (compensation, fault and injury characteristics) and pain severity, pain interference, health status and disability were examined with ordinal, linear and logistic regression, and mediation analyses.

**Results** Injury severity, compensable injury and external fault attribution were consistently associated with moderate-to-severe pain, higher pain interference, poorer health status and moderate-to-severe disability. The association between compensable injury, or external fault attribution, and disability and health outcomes was mediated via pain self-efficacy and perceived injustice.

**Conclusions** Given that the associations between compensable injury, pain and disability was attributable to lower self-efficacy and higher perceptions of injustice, interventions targeting the psychological impacts of pain and injury may be especially necessary to improve long-term injury outcomes.

### Strengths and limitations of this study

► Our understanding of the link between compensable injury and poor recovery has been limited by the varying nature of compensation system and systematic methodological factors, especially given that only those who are not at fault are eligible for compensation in many settings, and those with a poor recovery are more likely to lodge a claim.

► The regionalised Victorian trauma system, trauma registry platforms, and compensation system design (i.e., no fault) means that this setting is ideal to investigate compensable injury outcomes, and fault-related outcomes.

► While the present sample was large and represented a range of injury severities, the findings should be taken in light of the fact that the sample had relatively higher socioeconomic status than the Victorian injury population.

► This cross-sectional observational cohort study identified theoretically based sequential associations between compensable injury (and fault attribution), psychological appraisals of pain and/or injury and level of function and health status 1 year after injury.

## INTRODUCTION

Pain and injury are a leading contributors to global disease burden.[1] After traumatic injury,[2] disabling pain affects one in every three to four persons 3 years later,[3 4] making injury a significant cause of chronic pain in the community. Compensable injury, or the eligibility for and/or pursuit of an injury compensation claim, paradoxically leads to worse outcomes, including chronic and disabling pain.[5] This is despite the fact that compensation claimants are typically entitled to more benefits to support recovery, including healthcare and income replacement,[6–8] and some may receive lump sum payments depending on the setting.

Several factors may explain the 'compensation effect'. First, symptom exaggeration and

malingering are thought to be present to varying degrees in up to 30% of injury claimants.[9] Moreover, those who seek compensation may selectively represent those who have a worse outcome (or those who are more likely to report poorer outcomes, eg, seeking secondary gain[10]). It should also be noted that compensable injury typically involves more severe injury, especially multitrauma transport injury. Altogether, these factors often result in misleading 'reverse causality' explanations of the effect of compensation on recovery.[11] Nonetheless, even when studies account for injury characteristics, those who had a compensable injury are still often found to have worse recovery.[12]

Aside from methodological problems in the literature, key mechanisms through which compensable injury may result in poorer outcomes include the additive experience of stress from engaging with compensation systems (eg, due to perceived lack of power),[13] having to prove that another was at fault,[14] and the effects of procedural factors on perceived injustice and stress. Specific sources of procedural injustice include: (a) poor access to clear and timely information about compensation procedures or application outcomes, (b) perceived lack of empathy or engagement in interactions and (c) dissatisfaction with decisions about individual entitlements.[15 16]

While compensable injury is consistently associated with poorer long-term injury outcomes, the mechanistic role of psychological appraisals of pain and/or the injury have rarely been examined. A large body of work has demonstrated that pain catastrophising (defined as the tendency towards having an exaggerated or excessive focus on negative aspects of pain and a lack of control over pain[17]) is associated with the persistence of pain and disability.[18 19] Fear of exacerbating pain or causing re-injury (ie, kinesiophobia) and self-efficacy appraisals, which increase the likelihood of avoiding activity,[20 21] are also associated with worse disability,[22] and poorer quality of life .[23–26] Moreover, persistent pain and disability after compensable injury are associated with negative appraisals of compensation-related experiences,[16] which may co-occur with maladaptive cognitive appraisals of pain and perceptions of injustice.[27–31] In fact, the *belief* that another was at fault, or to blame, is consistently associated with worse outcomes after compensable injury,[32 33] especially in settings where determinations of fault are central to eligibility for compensation.[34] Altogether, injustice appraisals and stress after injury may increase the likelihood of transitioning from acute to chronic pain due to their concurrent impacts on behaviour and stress , which may disrupt the capacity to process, regulate and cope with painful sensations.[35 36]

While many studies have shown that compensable injury is associated with greater likelihood of developing persistent pain,[5] whether persons who sustain a compensable injury have worse pain because they also have maladaptive appraisals of pain is not known. This observational registry-based cohort study examined the experience of pain, catastrophising, kinesiophobia and self-efficacy after compensable and non-compensable injury, and examined the association between these psychological factors in the experience of pain, disability and health status 1 year after injury. We hypothesised that those with a compensable injury, and those who perceived that another was at fault, would be more likely to report severe and disabling pain, and that pain and disability) would be mediated by lower self-efficacy, and higher pain catastrophising, kinesiophobia and perceived injustice.

## METHODS

### Participants and recruitment

Participants were recruited from the Victorian State Trauma Outcomes Registry (VSTR) and the Victorian Orthopaedic Trauma Outcomes Registry (VOTOR),[37–39] 12 months after admission to hospital for traumatic injury. Only English-speaking participants aged 18–70 years were eligible to participate. Exclusion criteria were cognitive impairment as assessed qualitatively during trauma registry interview, participation in the registry via proxy representative, or high levels of distress. Distress was evaluated qualitatively by the registry interviewers, all of whom had worked in this role for several years, and was based on the participant's inability to complete the registry interview due to distress, or expressions of self-harm or suicidal ideation.

The VSTR and VOTOR registries are held in the Department of Epidemiology and Preventive Medicine, Monash University, and the same interviewers collect follow-up information for both registries. The registries comprise comprehensive details about patient demographics and injury and admission data, including trauma cause, mechanism and place, hospital admission, diagnoses and procedures. Injury and pain outcomes are assessed through telephone interviews at 6, 12 and 24 months following injury. The present study collected baseline and 12-month data from the registries, and administered additional questionnaires about pain, mental health and psychological factors related to the injury or pain (ie, catastrophising, kinesiophobia, self-efficacy and perceived injustice) 1 year after injury.

Participants are included in VSTR if they meet major trauma criteria, defined as (a) admission to the intensive care unit for >24 hours and mechanically ventilated; (b) significant injury to two or more body regions (ie, an Abbreviated Injury Scale (AIS)[6] scoring criteria) score of >2 in two or more body regions) or a total injury severity score (ISS) >12, or (c) urgent surgery for intracranial, intrathoracic or intra-abdominal injury, or fixation of pelvic or spinal fractures. Patients are included in VOTOR if they had orthopaedic (bone or soft tissue) injuries not related to metastatic disease, and were admitted to hospital for >24 hours. Patients are provided with information about the registries before the first follow-up interview, and are given the opportunity to opt-off. Less than 1% of patients elect to be removed from VOTOR or VSTR.

The present strategy to recruit from both VSTR and VOTOR aimed to ensure that (a) the cohort comprised a range of injury severity; (b) potential sources of bias could be identified through comparison with other publications of these registry patients and (c) reliance on patient recall or medical record review was minimised as injury and admission data were available from the registries.

## Materials and procedures

The study was approved by Alfred Hospital (study: 290/13) and Monash University (study: CF13/3276—2013001633) human research ethics committees. Participants were invited into the present study by trauma registry staff at the conclusion of the 12-month registry interview if they were treated at The Alfred Hospital, one of the two major trauma services in Victoria, Australia. Participants were not informed of the specific study hypotheses, but that the study was examining which factors affect recovery from traumatic injury. Participants were reassured that their data would not be shared with any other parties. All participants gave informed written consent to participate in this study, and for the researchers to obtain data from the trauma registries. Participants then completed additional questionnaires either via telephone interview, online or in hard copy.

## Demographics and preinjury health

Participant characteristics collected from the registries included sex, age at time of injury, education level and work status. Presence of comorbidities or other pre-existing health conditions at the time of hospital admission were determined using the International Statistical Classification of Diseases and Related Health Problems, Tenth Revision, Australian Modification diagnosis codes. Participants were also asked about other existing health conditions that might not have been captured at initial admission.

## Injury characteristics

Injury data extracted from the trauma registries included AIS 2005 Update,[40] ISS (the sum of the three most severe AIS scores, squared, from different body regions),[41] length of stay in hospital (in days) and discharge destination (ie, home or inpatient rehabilitation). In all cases, AIS scores were coded retrospectively by trained and experienced AIS coders either employed by the health service or the Victorian State Trauma Registry. The maximum AIS score across body regions (ie, head, neck, thorax, abdomen, spine, upper extremity, lower extremity, unspecified), and the number of body regions with an AIS score ≥2 (ie, moderate-to-critical injuries) were used to reflect injury severity, as ISS has previously been shown to have little to no association with pain after injury when adjusting for other demographic and injury covariates.[39] Trauma place (ie, transport, work, home or other), and whether or not the person felt they were at fault, were recorded.

The injury was defined as compensable if it was classified as such from the hospital records in VOTOR or VSTR, if the participant reported during our interviews that they had lodged a compensation claim (including victims of crime or public liability), or if the participant was eligible for compensation due to the setting and circumstances of their injury. That is, in Victoria, transport injury involving a motorised vehicle or a vehicle that operates on rails automatically qualifies for assistance from the Traffic Accident Commission (TAC), and injury while in the course of paid work is compensable by WorkSafe Victoria.

## Pain and functional outcomes (12 months)

*The Brief Pain Inventory (BPI)* was used to quantify pain severity and interference of pain with various aspects of daily life on 11-point Numeric Rating Scales[42] (Cronbach's α=0.92 for pain severity subscale and 0.95 for pain interference subscale in the present cohort). Scores >=4/10 were considered indicative of moderate-to-severe pain.[43 44]

Level of disability was measured using the extended version of the *Glasgow Outcome Scale (GOS-E),*[45] which classifies patient status into one of eight categories: death, vegetative state, lower severe disability, upper severe disability, lower moderate disability, upper moderate disability, lower good recovery and upper good recovery. Disability status was determined from independence, work and leisure activity participation, and relationships with family and friends, and classified as 'good' (ie, lower-upper good recovery) or moderate-to-severe disability (ie, vegetative state, lower severe disability, upper severe disability, lower moderate disability, upper moderate disability). The GOS-E has been shown to have good reliability and validity when using the structured interview format after head injury[45 46] and/or major trauma.[47]

The *EuroQol Five Dimensions Questionnaire (EQ-5D)*[48] was used to measure general health outcomes relating to five domains: mobility, self-care, usual activities, pain or discomfort and anxiety or depression. A summary score ranging from 0.00 to 1.00 was calculated using the UK indexed norms,[49] where a score of 1.00 indicates the best health state, and 0.00 indicates the worst health outcome. The UK tariffs were used as these are most commonly applied across international studies,[50] including previous Australian registry-based studies.[51] The EQ-5D shows sound validity and sensitivity to injury outcomes.[50 51]

## Psychological mediators (12 months)

The mediating effects of psychological characteristics related to pain were assessed by four measures: the Pain Catastrophising Scale (PCS), Pain Self-Efficacy Questionnaire (PSEQ), Injustice Experience Questionnaire (IEQ) and the Tampa Scale of Kinesiophobia (TSK).

The *PCS* measured the tendency to have an exaggerated negative mindset in response to painful experiences.[17] It comprises 13 items, and respondents rated the degree to which they had certain thoughts and feelings when in pain (from 0 'not at all' to 4 'all the time'). All items were

summed to create a total score (Cronbach's α=0.95 in the present sample).

The *PSEQ*[52] is a 10-item inventory assessing how confident a person was that they can cope with their pain and accomplish the activities of daily life despite their pain. Confidence in these abilities was rated on a scale from 0 'not at all confident' to 6 'completely confident', and items were summed to create a total score (Cronbach's α=0.96 in the present data).

The *IEQ*[53] is a 12-item questionnaire on which respondents indicate the frequency of certain thoughts from 0 'never' to 4 'all the time', reflecting blame or unfairness and irreparability of loss due to an injury, which are summed to create a total score (Cronbach's α=0.95 in the present data).

The *TSK*[54] is a 17-item self-report measure of kinesiophobia (ie, fear of movement or fear of re-injury from movement). A total score was calculated by summing all responses after inverting items 4, 8, 12 and 16 (Cronbach's α=0.84 in the present data).

## Data analytic approach

Data were analysed with Stata statistical software V.14.0[13] (StataCorp, College Station, Texas, USA). Significance was determined if α<0.05, or if the 95% CI did not include 1.00 (logistic and ordinal regression) or 0.00 (linear regression, mediation). Participants with missing data (<5.0% of cases across respective analyses) were excluded from the respective analysis in a list-wise manner. The data were summarised with descriptive statistics.

The design of the primary analyses is summarised in figure 1. Ordinal regression examined ordinal variables (ie, pain severity; 0=no pain,<4 =low pain, >=4=moderate-to-severe pain),[43 44] linear regression examined continuous variables (ie, pain interference and EQ-5D summary score), and logistic regression for binary variables (ie, GOS-E; 'good' recovery vs moderate-to-severe disability). Univariable regression models were fit to examine the relationship between each independent and dependent variable while controlling for age, sex, pain severity and injury severity (number of body regions with moderate-to-severe AIS score). Violation of the proportional odds assumption was assessed for ordinal models,

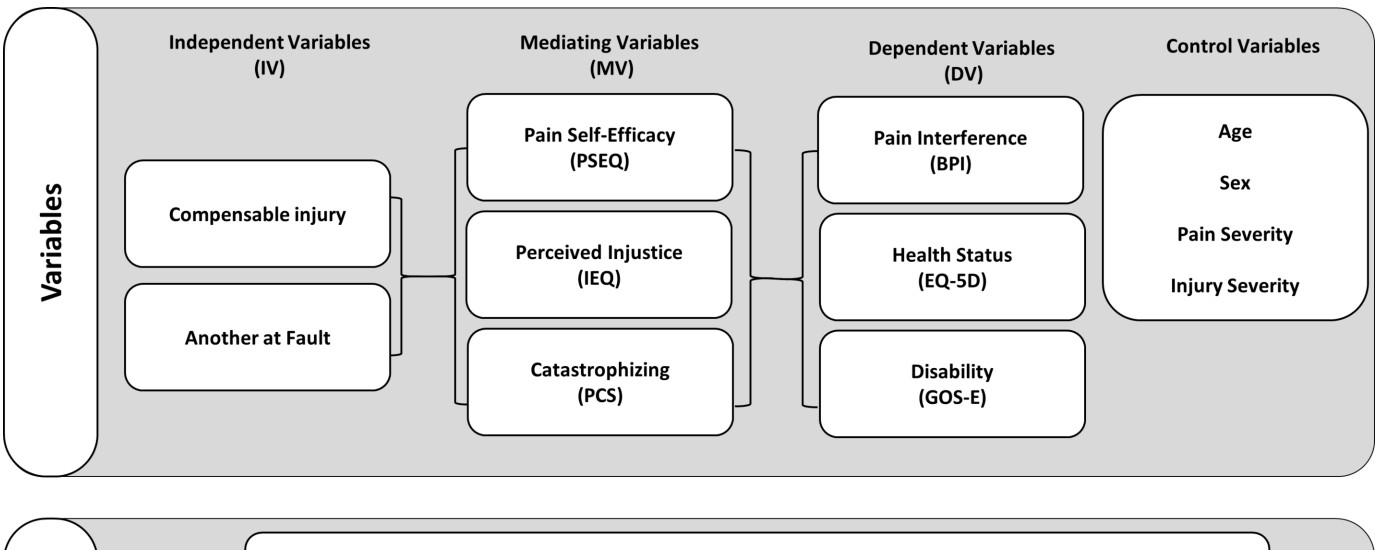

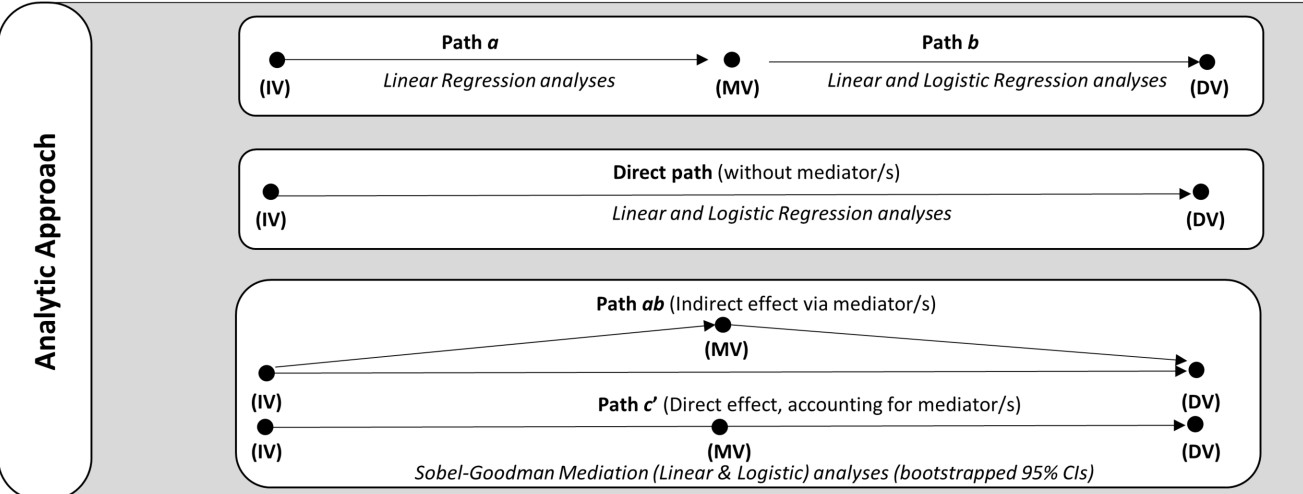

**Figure 1** Study and analysis design. BPI, Brief pain inventory; EQ-5D, EuroQol five dimensions questionnaire; GOS-E, extended version of the glasgow outcome scale; IEQ, Injustice experience questionnaire; PCS, Pain catastrophising scale; PSEQ, Pain self-efficacy questionnaire.

and effects were reported in accordance with these assumptions.

Mediation analyses examined the sequential relationship between the independent variables (compensation and fault status), via the mediating variables (self-efficacy, catastrophising and perceived injustice), and the dependent variables (pain interference, health status and disability). The strength of indirect effects was only examined if the proposed mediator was significantly associated with both the independent and dependent variables in preliminary linear and logistic regression analyses.[55] Mediated relationships were tested using the Sobel-Goodman mediation test with linear analyses for continuous factors (BPI interference, EQ-5D summary score) or logistic analyses for the categorical outcome (GOS-E), and bootstrapping with 500 case resamples. All mediation analyses adjusted for age, sex, injury severity (number of body regions with moderate-to-severe AIS score) and pain severity. The presence and strength of indirect, or mediated, effects were determined from examination of the size of the coefficient, and the bootstrapped 95% CIs such that effects were considered significant if the CI did not contain zero. The mediated effects were defined as 'partial mediation' if the direct effect (path c') was smaller and of the same sign as the indirect effect but remained significant, or as 'complete mediation' if the indirect effect equalled the total effect, and the direct effect (path c') was no longer significant.[55] Effect estimates were interpreted as *very small* (<0.01), *small* (≥0.20), *moderate* (≥0.50), *large* (≥0.80), *very large* (≥1.20) or *huge* (≥2.0).[56]

The sample (n=433) was sufficiently powered for the univariate linear and logistic regression conducted (with adjustment for four covariates: age, sex, pain severity, injury severity (ie, number of body regions with a moderate-to-severe AIS score), and for detection of moderate bias-corrected bootstrapped indirect effects, which require a minimum sample of 377 and 400 cases, respectively.[57]

## RESULTS
### Cohort overview
All participants were admitted to hospital after traumatic injury from October 2012 to October 2014. A total of 732 patients were referred to the study during their 12-month VOTOR or VSTR registry interview. Seventy potential participants could not be contacted leaving 662 assessed for eligibility. Twelve participants were ineligible (two were deceased, seven were distressed and three were unwell), and 217 declined to participate resulting in a sample of 433 participants (66.6% response rate).

The average time from injury to follow-up was 13.50 months (SD=1.60 months). The participants were predominately male (74.8%), average age at time of injury was 44.8 years (SD=14.2) and the majority of participants had completed postsecondary school education (63.9%), which is slightly higher than the general

Australian population of 61% of persons aged 15–64 years have a postschool qualification.[58] Almost two-thirds had a household income >$A60 000 per annum (60.9%) 12 months after injury, which is slightly higher than the national average household income of $A52,000.[59] One hundred and sixty-nine participants had a compensable injury, including a transport-related injury (n=141) or workplace injuries (n=28). See table 1 for an overview of the cohort characteristics (n=433).

Two hundred and sixty-seven (61.7%) patients were registered to both VSTR and VOTOR, 111 (25.6%) patients were in VOTOR only, and 55 patients were in VSTR only. Consistent with the registry inclusion criteria, participants recruited from VSTR had higher ISS (and maximum AIS) than participants recruited from VOTOR only (ISS: mean difference=9.39, 95% CI 7.45 to 11.32; maximum AIS: mean difference=0.89, 95% CI 76 to 1.03). There was no difference between participants who were registered to VSTR compared with VOTOR on reported pain severity (mean difference=−0.016, 95% CI −0.46 to 0.43), pain interference (mean difference=0.22, 95% CI −0.29 to 0.74), pain catastrophising (mean difference=1.01, 95% CI −1.38 to 3.40), kinesiophobia (mean difference=0.21, 95% CI −0.1.57 to 1.99) or health status (mean difference=0.013, 95% CI −0.38 to 0.063) 1 year after injury. Participants recruited from VSTR had lower pain self-efficacy (mean difference=−3.60, 95% CI −6.71 to −0.50), and higher perceived injustice (mean difference=4.09, 95% CI 1.15 to 7.03), and were also more likely to have moderate-to-severe disability 1 year after injury than those only registered to VOTOR (Risk Ratio (RR): 1.49, p=0.003). This latter difference is expected given that permanent disability (eg, due to cognitive, functional, social or psychological impairments) is more likely to arise after major trauma than orthopaedic trauma.

### Factors associated with pain severity
Data on pain and pain-related outcomes are summarised in table 2. At 1 year postinjury, the majority of participants reported pain of low severity (ie, <4/10; n=258, 59.6%), 63 (14.5%) reported no pain at all and 112(25.9%) reported moderate-to-severe pain (ie, >=4/10). A relatively small proportion of participants had clinically significant scores across measures, including pain interference (scores >=4; n=120, 27.8%), catastrophising (scores >=30; n=34, 7.9%), self-efficacy (scores <20; n=26, 6.1%), kinesiophobia (scores >40; n=172, 39.9%) and perceived injustice (scores >20; n=159, 36.9%).

There was a modest correlation between age and pain severity ($r_s$=0.13, p<0.006), and females were more likely to report moderate-to-severe pain than males (OR 1.71; 95% CI 1.06 to 2.75). Participants with lower education (ie, year 11 or below) were more likely to report moderate-to-severe pain (OR 2.80; 95% CI 1.55 to 5.05) than those with postsecondary education. Likewise, participants who were not employed prior to injury (RR 3.35, 95% CI 1.65 to 6.81), or had not returned to work 12

**Table 1** Cohort characteristics

| | Category | Total | | Compensable n=169 | | Not compensable n=264 | |
|---|---|---|---|---|---|---|---|
| | | n | % | n | % | n | % |
| **Demographic characteristics** | | | | | | | |
| Sex | Male | 324 | 74.8 | 128 | 75.7 | 196 | 74.2 |
| | Female | 109 | 25.2 | 41 | 24.3 | 68 | 25.8 |
| Age (years) at injury | 18–30 | 91 | 21.3 | 41 | 24.7 | 50 | 19.1 |
| | 31–40 | 70 | 16.4 | 23 | 13.9 | 47 | 17.9 |
| | 41–50 | 82 | 19.2 | 38 | 22.9 | 44 | 16.8 |
| | 51–60 | 122 | 28.5 | 42 | 25.3 | 80 | 30.5 |
| | 61+ | 63 | 14.7 | 22 | 13.3 | 41 | 15.6 |
| Presence of >1 comorbidity | None | 274 | 63.3 | 109 | 64.5 | 165 | 62.5 |
| | ≥1 | 159 | 36.7 | 60 | 35.5 | 99 | 37.5 |
| Highest education | Postsecondary education* | 272 | 64.5 | 102 | 63.4 | 170 | 65.1 |
| | Completed year 12 | 64 | 15.2 | 27 | 16.8 | 37 | 14.2 |
| | Year 11 or less | 86 | 20.4 | 32 | 19.9 | 54 | 20.7 |
| Household income (p/a at 12 months after injury) | $A20–40 000 | 98 | 23.6 | 40 | 26.3 | 58 | 22.1 |
| | $A41–60 000 | 64 | 15.4 | 23 | 15.1 | 41 | 15.6 |
| | $A61–80 000 | 67 | 16.1 | 30 | 19.7 | 37 | 14.1 |
| | $A81–1 00 000 | 51 | 12.3 | 19 | 12.5 | 32 | 12.2 |
| | $A100 000+ | 135 | 32.5 | 40 | 26.3 | 95 | 36.1 |
| **Work characteristics** | | | | | | | |
| Employment field | White collar | 179 | 41.3 | 56 | 35.0 | 123 | 45.1 |
| | Blue collar | 174 | 40.2 | 76 | 47.5 | 98 | 35.9 |
| | Not working/studying | 80 | 18.4 | 28 | 17.5 | 52 | 19.1 |
| **Injury characteristics** | | | | | | | |
| Moderate-critical injury† | 1. Head | 120 | 27.7 | 59 | 34.9 | 61 | 23.1 |
| | 2. Face | 82 | 18.9 | 42 | 24.9 | 40 | 15.2 |
| | 3. Neck | 12 | 2.8 | 8 | 4.7 | 4 | 1.5 |
| | 4. Thorax | 146 | 33.7 | 90 | 53.3 | 56 | 21.2 |
| | 5. Abdomen | 50 | 11.5 | 39 | 23.1 | 11 | 4.2 |
| | 6. Spine | 151 | 34.9 | 65 | 38.5 | 86 | 32.6 |
| | 7. Upper extremity | 165 | 38.1 | 77 | 45.6 | 88 | 33.3 |
| | 8. Lower extremity | 218 | 50.3 | 100 | 59.2 | 118 | 44.7 |
| | 9. Unspecified | 35 | 8.1 | 16 | 9.5 | 19 | 7.2 |
| Discharge destination | Home | 304 | 70.2 | 92 | 54.4 | 212.0 | 80.3 |
| | Rehabilitation | 129 | 29.8 | 77 | 45.6 | 52.0 | 19.7 |

*Postsecondary education included postsecondary school certificate, diploma, bachelor or postgraduate degree.
†Body region with severe injury with an AIS severity score of 2–5, and multiple body regions could be affected for each participant.
AIS, Abbreviated Injury Scale; p/a, per annum.

months postinjury (OR 2.95, 95% CI 1.69 to 5.15), were more likely to report moderate-to-severe pain than those who were working before injury or had returned to work, respectively.

Relationships between baseline injury characteristics and pain severity 1 year postinjury are reported in table 3. Participants were more likely to have pain if they had a more severe injury, such that for each additional body region with a moderate-to-critically severe injury, there was a 37% increase in the odds of having moderate-to-severe pain 1 year after injury. The likelihood of having moderate-to-severe pain was also associated with a longer hospital stay (4% increased odds of worse pain for each additional day), having a compensable injury (32%

**Table 2** Pain and pain-related characteristics in compensable and non-compensable participants

| | Measure | Statistic | Compensable n=160 | Not compensable n=273 | p Value | Effect size |
|---|---|---|---|---|---|---|
| Pain severity | BPI | M (SD) | 2.94 (2.19) | 2.30 (1.94) | 0.002 | 0.31 |
| Pain interference | BPI | M (SD) | 3.39 (2.78) | 2.16 (2.28) | <0.001 | 0.48 |
| Pain catastrophising | PCS | Md (IQR) | 8.00 (16.00) | 4.00 (13.00) | <0.001* | 0.17 |
| Pain self-efficacy | PSEQ | M (SD) | 41.41 (15.43) | 47.78 (13.14) | <0.001 | 0.44 |
| Kinesiophobia | TSK | M (SD) | 38.45 (8.39) | 36.30 (7.99) | 0.008 | 0.26 |
| Perceived injustice | IEQ | M (SD) | 20.52 (14.61) | 13.73 (12.40) | <0.001 | 0.50 |

Statistics were all independent samples t-tests, and Cohen's D effect sizes, except for pain catrastophising, which was examined with a non-parametric Mann-Whitney U test (and effect size calculation of z/√N).
BPI, Brief Pain Inventory; IEQ, Injustice Experience Questionnaire; M, mean; Md, median; PCS, Pain Catastrophising Scale; PSEQ, Pain Self-Efficacy Questionnaire; TSK, Tampa Scale of Kinesiophobia.

increased odds of pain) and attributing fault to another (46% increased odds of pain). However, place of injury (ie, transport, work, home or elsewhere), compensation status and fault attribution were not related to pain severity when adjusting for all injury and demographic characteristics.

**Factors associated with psychological variables (*a* paths)**
Figure 2 shows associations between baseline injury characteristics and psychological functioning in relation to pain at 12 months (adjusting for age, sex, pain severity and injury severity). Catastrophising, self-efficacy and perceived injustice were all worse in those who were discharged to inpatient rehabilitation following their injury, and in those who attributed fault to another. Self-efficacy was lower in participants who had a compensable injury or a longer

**Table 3** Relationship between injury characteristics and pain severity (ordinal regression)

| Characteristics | | No pain n=63 (14.5%) | Low pain n=258 (59.6%) | Moderate-to-severe pain n=112 (25.9%) | OR | OR^adj (95% CI) |
|---|---|---|---|---|---|---|
| Injury severity | | | | | | |
| AIS count† | M (SD) | 1.51 (0.82) | 1.75 (1.07) | 2.13 (1.33) | 1.38 | 1.37 (1.15, 1.62)* |
| Hospital stay (continuous)‡ | | | | | | |
| None vs any pain | M (SD) | 6.49 (6.36) | 6.72 (8.13) | | 1.00 | 0.98 (0.95 to 1.02) |
| None/low vs moderate/severe pain | M (SD) | 5.69 (5.99) | | 9.53 (11.31) | 1.05* | 1.04 (1.01, 1.07)* |
| Injury place | | | | | | |
| At home | N (%) | 14 (22.2) | 45 (17.4) | 18 (16.1) | Ref | Ref |
| Traffic/road | N (%) | 23 (36.5) | 96 (37.2) | 54 (48.2) | 1.52 | 1.38 (0.75 to 2.52) |
| Workplace | N (%) | 4 (6.4) | 25 (9.7) | 16 (14.3) | 1.98 | 1.99 (0.93 to 4.26) |
| Other | N (%) | 22 (34.9) | 92 (35.7) | 24 (21.4) | 0.88 | 1.11 (0.60 to 2.06) |
| Compensation status | | | | | | |
| None | N (%) | 41 (65.1) | 169 (65.5) | 55 (49.1) | Ref | Ref |
| Traffic Accident Commission/ worksafe | N (%) | 22 (34.9) | 89 (34.5) | 57 (50.9) | 1.68* | 1.32 (0.84 to 2.07) |
| Fault | | | | | | |
| At fault | N (%) | 36 (57.1) | 133 (52.0) | 46 (41.8) | Ref | Ref |
| Not at fault | N (%) | 27 (42.9) | 123 (48.0) | 64 (58.2) | 1.50* | 1.46 (0.99 to 2.15) |

Significant relationships are with an asterisk (*). OR^adj have adjusted for age, sex and education. Analysis of hospital stay, injury place, compensation, fault and work status also controlled for injury severity (number of body regions with moderate-to-severe AIS score).
†AIS count=the number of moderate-to-critical injured body regions.
‡The proportional odds assumption was not met for length of hospital stay, so ORs are reported here for each ordinal comparison.
AIS, Abbreviated Injury Scale.

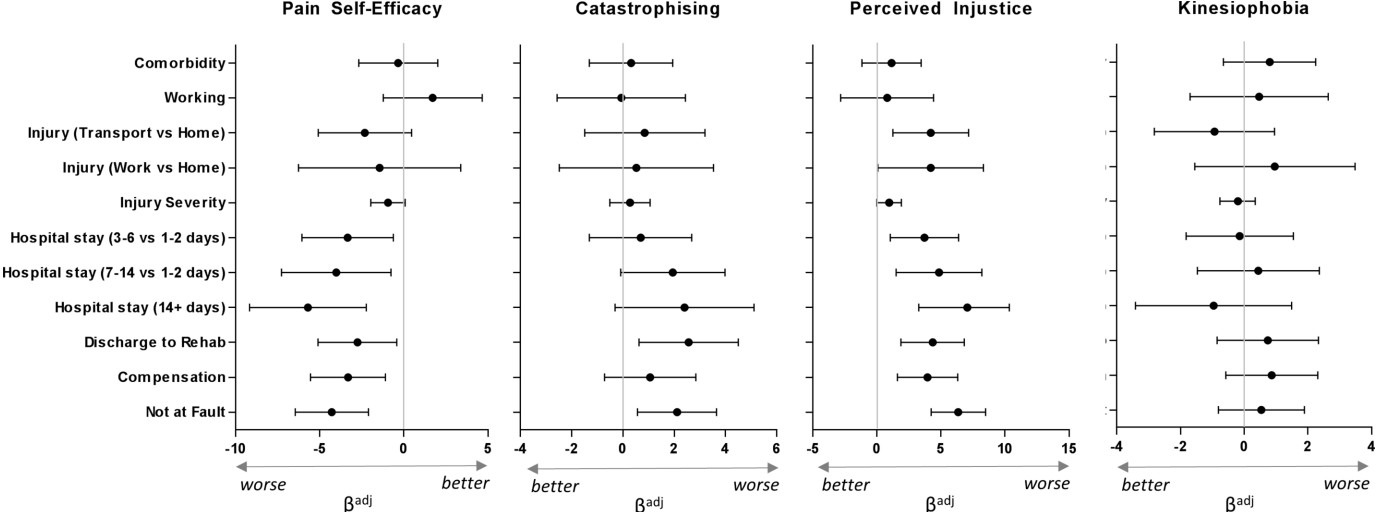

**Figure 2** Regression beta weights and ORs for the association between injury characteristics and psychological variables of pain catastrophising, pain self-efficacy, kinesiophobia and injustice experience, adjusted for age, sex and injury severity (path a). Error bars (95% CI) that do not cross the central line indicate significant relationships. Tables of specific values can be found in online supplementary material.

hospital stay. Perceived injustice was worst in participants with transport or work-related injuries compared with those with an injury at home or elsewhere, in participants with compensable injury and with longer hospital stay. Kinesiophobia was not related to any injury characteristics.

### Factors associated with poor functional recovery (direct effects and path b)

Examination of the direct effects of injury characteristics and psychological responses to the pain or injury on pain interference were examined only in participants who reported some pain 12 months after injury (n=370), see figure 3. Most

participants reported low levels of pain interference (<4/10; n=249, 67.5%), and the remainder (n=120, 32.3%) reported moderate-to-severe pain interference (ie, ≥4/10), with average pain interference of 3.04 (SD=2.49) in participants reporting some pain.

The relationships between injury characteristics and psychological responses to the pain or injury and the EQ-5D and GOS-E were examined in all participants. The average EQ-5D summary score was 0.80 (SD=0.23), indicating moderate-to-good health status in the majority of participants. According to the GOS-E, 210 (48.5%) participants

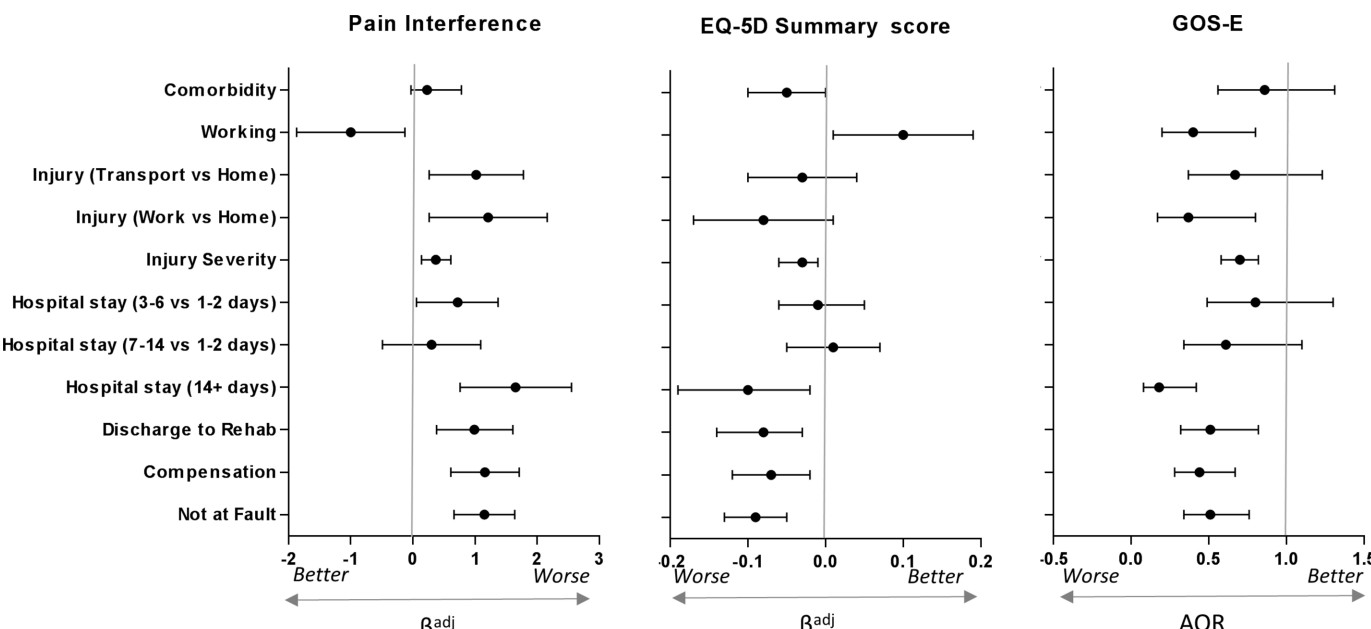

**Figure 3** Regression for association between baseline characteristics and functional recovery outcomes of pain interference (only for those with pain severity >0; n=370), EuroQol five dimensions questionnaire (EQ-5D) and Extended version of the glasgow outcome scale (GOS-E), adjusted for age, sex and injury severity. Error bars (95% CI) that do not cross the central line indicate significant relationships.

**Table 4** Association between mediators and pain interference, health status (EQ-5D summary score) and GOS-E recovery (path b)

| Mediators | Pain interference | | EQ-5D Summary score | | GOS-E Functional outcome | |
|---|---|---|---|---|---|---|
| | Β | (95% CI) | β | (95% CI) | OR | (95% CI) |
| Pain severity | 1.02 | 0.95 to 1.10 | −0.065 | −0.077 to 0.053 | 0.59 | 0.52 to 0.68 |
| Pain self-efficacy | −0.06 | −0.08 to 0.04 | 0.005 | 0.003 to 0.007 | 1.04 | 1.01 to 1.06 |
| Kinesiophobia | 0.07 | 0.04 to 0.09 | −0.004 | −0.007 to 0.001 | 0.95 | 0.91 to 0.98 |
| Catastrophising | 0.08 | 0.06 to 0.10 | −0.007 | −0.010 to 0.004 | 0.95 | 0.91 to 0.98 |
| Perceived injustice | 0.06 | 0.04 to 0.07 | −0.004 | −0.006 to 0.003 | 0.94 | 0.92 to 0.96 |

Pain interference and EQ-5D summary score were analysed with linear regression, GOS-E was analysed with logistic regression (comparing 'good' recovery vs moderate-to-severe disability, where higher odds indicate increased likelihood of the good recovery). All analysis adjusted only for age, sex, pain severity and injury severity (number of body regions with moderate-to-severe AIS score). The sample for pain interference regression only comprised participants reporting a pain intensity >0; n=370).
AIS, Abbreviated Injury Scale; EQ-5D, EuroQol Five Dimensions Questionnaire; GOS-E, extended version of the Glasgow Outcome Scale.

had 'good' functional recovery, 216 (49.9%) had moderate disability and 7 (1.6%) had severe disability. Given the small number of patients who had severe disability, the moderate and severe disability groups were combined for all analyses.

Figure 3 shows the relationship between baseline characteristics and the functional and health-related dependent variables. Participants had poorer health or function across all three measures (BPI interference; EQ-5D; GOS-E) if they sustained a compensable injury, attributed fault to another or required inpatient rehabilitation (see online supplementary material for specific ORs and CIs). One-year postinjury, disability (GOS-E) was more likely in those who were *employed* prior to injury, whereas pain interference and health status were worse in those who were *unemployed* prior to injury. Pre-existing medical conditions were not associated with any dependent variable; however, it should be noted that the sample generally had good health prior to injury, with only 132 (35.7%) patients reporting one or more comorbidities at the time of the injury, and the cohort had an average rating of preinjury health of 89.16 (SD=10.76) out of 100, where 100 indicates 'best imaginable health state'.

All psychological variables (self-efficacy, kinesiophobia, catastrophising and perceived injustice) were predictive of higher pain interference, lower health status and disability outcome after controlling for demographics, pain severity and injury severity (table 4).

### Indirect effects on functional outcomes (ab path; path c')

As kinesiophobia was not associated with compensation or fault, and catastrophising was not associated with compensation, these variables were not included as potential mediators in the respective analyses. The effect estimates, and bootstrapped 95% CIs, of the indirect and direct effects are shown in table 5.

### Pain interference

There was partial mediation between compensation and pain interference via pain self-efficacy and perceived injustice, and between fault and pain interference via pain self-efficacy, perceived injustice and catastrophising. The size of the direct and indirect effects were small to moderate, with the combined effect of the mediators explaining 59.3% and 48.7% of the total variance in the association between compensation, or fault attribution, and pain interference, respectively.

### Health

There was complete mediation of the relationship between compensation and health status via self-efficacy and perceived injustice. There was also complete mediation of the relationship between fault and health status, and disability, via perceived injustice, and only partial mediation between fault and health status via self-efficacy and catastrophising. While the magnitude of both the direct and indirect effects between compensation and fault on health status would be considered very small, with all estimates being <0.04, it should be noted that 54.0% and 50.1%, respectively, of the total variance in the association was indirect via self-efficacy, perceived injustice and catastrophising.

### Disability

There was partial mediation between compensation and disability via perceived injustice (of moderate effect size), but no mediation via self-efficacy. There was complete mediation between fault and disability via perceived injustice (of small effect size), partial mediation via self-efficacy (moderate effect size) and no mediation via catastrophising. The combined indirect effects, via the mediators, explained 55.6% and 25.1% of the total variance in the association between compensation, or fault attribution, and disability, respectively.

### DISCUSSION

This study demonstrates that characteristics at the time of injury, especially compensable injury and attributing fault to another, were consistently associated with poorer health and level of function 1 year after injury, including

**Table 5** Direct (path c') and indirect (mediated, path ab) effects, with bootstrapped 95% CI, between independent variables (compensation; fault) and functional or health status (BPI interference; EQ-5D; GOS-E), mediated by psychological characteristics (pain self-efficacy; perceived injustice; pain catastrophising)

| | Pain interference (BPI) | | Health (EQ-5D) | | Disability (GOS-E) | |
|---|---|---|---|---|---|---|
| | Path *ab* indirect effect path *c'* | Direct effect | Indirect (mediated) effect | Direct effect* | Indirect (mediated) effect | Direct effect* |
| | *B* (95% CI) | *B* (95% CI) | *B* (95% CI) | *B* (95% CI) | *B* (95% CI) | *B* (95% CI) |
| Pain self-efficacy | | | | | | |
| Compensation | 0.24 (0.07 to 0.41) | 0.41(0.08 to 0.75) | −0.02 (−0.29 to 0.002) | −0.03(−0.07 to 0.009) | −0.25(−0.06 to 0.01) | −0.19 (−0.31 to 0.07) |
| Fault | 0.25 (0.0 to 90.41) | 0.42 (0.13 to 0.72) | −0.02(−0.35 to 0.004) | −0.04(−0.08 to 0.007) | −0.32(−0.06 to 0.001) | −0.13 (−0.25 to 0.01) |
| Perceived injustice | | | | | | |
| Compensation | 0.25 (0.10 to 0.40) | 0.39(0.08 to 0.70) | −0.02(−0.03 to 0.01) | −0.03(−0.07 to 0.01) | −0.05(−0.09 to 0.01) | −0.16(−0.28 to 0.05) |
| Fault | 0.34 (0.19 to 0.48) | 0.33(0.02 to 0.63) | −0.03(−0.04 to 0.02) | −0.03 (−0.06 to 0.004) | −0.08(−0.13 to 0.04) | −0.08(−0.20 to 0.04) |
| Catastrophising | | | | | | |
| Fault | 0.17 (0.03 to 0.32) | 0.49 (0.18 to 0.77) | −0.01 (−0.03 to 0.002) | −0.04(−0.07 to 0.01) | −0.03(−0.05 to 0.002) | −0.13(−0.25 to 0.01) |
| Combined mediated effects | | | | | | |
| Compensation | 0.07 (0.03 to 0.10) | 0.06(0.006 to 0.13) | −0.05 (−0.08 to 0.02) | −0.05(−0.13 to 0.03) | −0.05(−0.09 to 0.02) | −0.16(−0.27 to 0.05) |
| Fault | 0.08 (0.04 to 0.12) | 0.05(−0.0005 to 0.11) | −0.07 (−0.10, to 0.03) | −0.06(−0.13 to 0.01) | −0.09(−0.13 to 0.05) | −0.07(−0.19 to 0.05) |

Analyses were univariate, adjusting for age, sex, pain severity and injury severity (number of body regions with moderate-severe AIS score).
*Direct effect while accounting for the effects of the mediator, and change in the magnitude of the direct effect should refer to the effects shown in figure 2. Effects are considered statistically significant if the 95% CI does not contain zero.
AIS, Abbreviated Injury Scale; BPI, Brief Pain Inventory; EQ-5D, EuroQol Five Dimensions Questionnaire; GOS-E, extended version of the Glasgow Outcome Scale.

pain-related disability. These associations were observed both before and after controlling for injury severity and demographic factors that were also associated with worse outcomes. A notable exception was that pain *severity* 1 year after compensable injury appeared to be most strongly associated with injury severity. So while this study has replicated the so-called 'compensation effect',[5 6 60] we suggest that the mechanism of injury (ie, transport crashes) for the majority of the compensable cases in this study may have increased the likelihood of having moderate-to-severe pain. Transport-related injuries tend to involve high energy collisions, and result in complex multitrauma, which are more likely to lead to persistent pain. Despite the clear association between injury severity and pain, we have shown for the first time that lower self-efficacy and higher perceptions of injustice after compensable injury mediated the degree to which pain impacted on a range of daily activities, as well as health and disability outcomes at 1 year after injury.

The total effect of fault perceptions on disability was found to be indirect via perceived injustice. While each of these factors no doubt covary after injury, and the sequential relationships could really be examined in varying combinations (eg, disability after injury also leads to perceptions of injustice), these findings are consistent with the frequent finding that external attributions of fault are associated with a range of poor health outcomes.[51 61] Here we show that when adjusting for injury severity, both attributions of fault and global perceptions of injustice are associated with reduced likelihood of having a good functional outcome. In other contexts, perceptions of injustice have been shown to have real and significant effects on rehabilitation outcomes, highlighting that disability is associated with a range of factors beyond the physical and functional limitations imposed by the injury. In fact, the harmful effects of perceived injustice have been shown to begin relatively early in the disability trajectory,[62] affect the quality of working relationship with health professionals,[63] promote behaviours that are not conducive to recovery[32] and lead to an inflexible focus on justice violations that may ultimately impede recovery.[33] In the worst-case scenario, injustice appraisals may even lead to chronic embitterment and a range of long-term mental health impacts, including depression and suicidal ideation.[64] Clearly, therefore, it is important to address injustice perceptions after injury, promote rehabilitation gains as early as possible, and attenuate any extrinsic contributors that exacerbate injustice perceptions (ie, procedural injustice).

This study demonstrated an association between compensable injury and worse disability and health outcomes (ie, in relation to mobility, self-care, activity participation, pain and anxiety/depression) that were, to varying degrees, attributable to the experience of lower self-efficacy and higher injustice perceptions. These findings suggest that patients who had a compensable injury, compared with those who sustained a non-compensable injury, were more likely to lack confidence in participating in activities of daily living because of persistent pain. Further to the impacts of injustice perceptions on behaviour, described above, low self-efficacy is known to increase the likelihood of adopting maladaptive behaviours and thoughts, such as fear avoidance and reduced participation in work, social and physical activities.[65] While self-efficacy did not mediate the relationship between compensable injury and disability, given that compensable injury leads to low pain self-efficacy it may be that it will lead to greater disability beyond this time frame.[23] Promoting self-efficacy, especially after compensable injury, is therefore a high priority for optimising health status and reducing pain interference after injury.[25]

While pain catastrophising was not worse after compensable injury, it did show a small association with pain interference, and it partially mediated the relationship between fault and pain interference, and health status. The effects via catastrophisng were very small across all analyses, which is most likely due to the fact that catastrophising characteristics were low in this cohort, with more than 90% of participants had catastrophising scores below the clinical threshold. However, it should be noted that this proportion is twice as high as that seen 6 months after musculoskeletal injury.[66] In the present study, only a quarter of the sample had moderate-to-severe pain, but just over half had moderate-to-severe disability. Therefore, we suggest that the unjust impacts of the injury, as measured by the IEQ,[31 67] may have been more pertinent in this cohort than pain-related catastrophising, as measured by the PCS.

Although kinesiophobia was associated with worse functional outcomes, it was not associated with any injury characteristics, including compensation or fault attributions. Evidently fear of re-injury, or exacerbating pain, is not closely associated with the severity of the initial injury. Rather, we speculate that emerging functional and psychological impacts of the injury, together with enduring personality traits, may play a greater role in kinesiophobia than injury severity.

## Clinical implications
It is clear that some injury and demographic characteristics increase the risk of persistent pain and disability after injury. There remains a pressing need now to develop and test effective psychosocial and medical interventions during the first year after injury to further improve long-term outcomes. Efforts should focus on modifying compensation procedures that exacerbate pain or psychological outcomes, and supporting recovery in those who believe that another was at fault, whether or not that belief is accurate.[51] Given that self-efficacy and perceived injustice showed important direct and indirect associations with function and health, further investigation is needed to understand whether these appraisals can be modified when specifically targeted in interventions.

At this stage, research on early interventions for the prevention of pain, disability and injustice beliefs after

injury is sparse.[33 68] Interventions delivered in the acute or subacute stage after injury that have been shown to have positive effects on self-efficacy typically comprise education,[69] and work towards building 'mastery' of activities that had become difficult because of pain. These interventions use behavioural achievements as a catalyst for positive change (ie, improved functional outcomes), which is more powerful than verbal encouragement alone.[25] Disability and perceptions of injustice are clearly bidirectionally associated.[33 53] Interventions targeting either injustice beliefs or functional restoration appear to elicit positive effects on the other.[70] New interventions could be developed and trialled to modulate injustice beliefs directly, especially for persons with injuries that result in permanent disability (eg, after spinal cord injury or brain injury). While injured persons may have very valid grounds for their beliefs, they may nonetheless benefit from therapies that enhance emotion control, acceptance[71] or forgiveness.[72] Ultimately, when designing any intervention to target complex psychological, pain and disability outcomes after injury, it is important to bear in mind that feelings of injustice frequently extend far beyond the person at fault for causing the injury, and may be directed toward the compensation system, employers, healthcare providers, lawyers and society as a whole.[62 73 74] It is therefore important that therapists and policy makers take a whole of person, and whole of system, approach to supporting injury recovery. Finally, procedures involved in claiming compensation, such as receiving timely and sufficient information, or having empathic interactions with claims staff, were not evaluated in this study, but should be evaluated to ensure that these procedures are not causing secondary harm.[15 73] Ultimately compensation systems are in a valuable position whereby they can optimise their systems and client relationships to bolster client self-efficacy.

### Strengths, limitations and future directions

The strengths and limitations of the present study should be considered when applying these findings to the trauma population. First, in the State of Victoria, all persons who are injured in transport (ie, involving a motorised or rail-operated vehicle) or workplace injuries are eligible for compensation, regardless of their role in the injury incident (ie, these are 'no fault' systems). When hospitalised, most cases will almost automatically have a claim number generated to reduce the need for a client to lodge their claim given that the medical excess has been met. This setting is therefore ideal for the examination of outcomes related to fault attributions, and compensable injury. That said, it should be noted that the present cohort, relative to the general population and trauma population studies in Victoria, had a relatively higher socioeconomic status (ie, slightly higher proportion of patients with postsecondary education[58] and annual income[59] than the national averages).

All of the mediating and dependent variables were measured 1 year postinjury. Although the present analyses

were theoretically driven (ie, given that *beliefs* about pain, or capacity to participate in activity, are predictive of actual behaviour and disability), these characteristics are known to frequently covary. Therefore, causal associations between psychological appraisals of pain or the injury (ie, catastrophising, self-efficacy, kinesiophobia and perceived injustice) and pain interference, health status and disability are not assumed, and further research is required to confirm these sequential associations, and their potential for change through intervention. Despite the cross-sectional nature of the study, our findings highlight that self-efficacy and perceived injustice are powerful indicators of poor injury outcomes alongside (or perhaps more so than) injury severity, and should be considered during injury rehabilitation.

In conclusion, while pain is more likely after compensable injury, this is largely because these injuries are more severe and complex. When accounting for injury severity, however, compensable injury was nonetheless found to lead to worse self-efficacy and health status, and higher perceived injustice, pain-related disability (pain interference) and disability. As perceived injustice and low self-efficacy played a key role in pain, health and disability after compensable injury, these characteristics warrant further investigation as a risk factor for pain and disability, and as targets for intervention.

**Author affiliations**
[1]Department of Epidemiology and Preventive Medicine, School of Public Health & Preventive Medicine, Monash University, Melbourne, Victoria, Australia
[2]Institute for Safety, Compensation & Recovery Research, Monash University, Melbourne, Victoria, Australia
[3]Caulfield Pain Management & Research Centre, Caulfield Hospital, Caulfield, Victoria, Australia
[4]School of Psychological Sciences and Monash Institute of Cognitive and Clinical Neurosciences, Monash University, Faculty of Medicine, Nursing and Health Sciences, Clayton, Victoria, Australia
[5]Academic Board of Anaesthesia & Perioperative Medicine, School of Medicine Nursing & HealthSciences, Monash University, Clayton, Victoria, Australia
[6]Monash-Epworth Rehabilitation Research Centre, Epworth Hospital, Richmond, Victoria, Australia

**Twitter** @MelitaGiummarra @EmergTrauma

**Acknowledgements** The authors acknowledge the contribution of Nellie Georgiou-Karistianis and Paul Jennings in the general design of the study, Melissa Hart and Mimi Morgan for participant recruitment, Susan McLellan for data extraction and injury coding, Amy Allen, Jessica Frisina and Samantha Finan for data collection.

**Contributors** Each coauthor contributed significantly to the work described and commented on the manuscript. MJG designed and supervised the conduct of the study, contributed to data preparation and analysis, discussed the results and co-wrote the manuscript with guidance and feedback from the coauthors. KSB assisted with patient recruitment and data collection, preparation of data for analysis, interpretation and writing of the manuscript. LI assisted with patient recruitment and data collection, preparation of data for analysis, interpretation and writing of the manuscript. SMG assisted with data analysis, interpretation of findings and writing and editing of the manuscript. SJG, CAA, JP and PC assisted in designing the project and analysis, interpretation of the findings and review of the manuscript.

**Funding** This research was supported by an Australian Research Council (ARC) Linkage Grant (LP120200033) in collaboration with the Victorian Transport Accident Commission (TAC), and a Platform Access Grant from Monash University (PAG15-0026). MJG was supported by an NHMRC early career fellowship (APP1036124), and PC was supported by a NHMRC fellowship (APP545926). The Victorian Orthopaedic Trauma Outcomes Registry (VOTOR) is funded by the Transport

Accident Commission via the Institute for Safety, Compensation and Recovery Research (ISCRR). The Victorian State Trauma Registry (VSTR) is a Department of Health, State Government of Victoria and Transport Accident Commission funded project. The Victorian State Trauma Outcome Registry and Monitoring (VSTORM) group is thanked for the provision of VSTR data. The funders were consulted throughout the design of the study, but played no role in data collection and analysis, the decision of how and what to publish or the preparation of this manuscript.

**Competing interests** None declared.

**Patient consent** Patients provided written informed consent.

**Ethics approval** Alfred Hospital and Monash University.

**Provenance and peer review** Not commissioned; externally peer reviewed.

**Data sharing statement** The Victorian State Trauma Registry (VSTR) and the Victorian Orthopaedic Trauma Outcomes Registry (VOTOR) have strict data/publication policies and guidelines to protect again unauthorised use or misinterpretation of the data. Data are only available on request from the corresponding author (MJG).

**Author note** MJG and KSB contributed equally to the paper, and are co-first author.

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
