## [Reviewer comments · BMJ Open]

ARTICLE DETAILS

TITLE (PROVISIONAL)	Associations between compensable injury, perceived fault and pain and disability one year after injury: A registry-based Australian cohort study
AUTHORS	Giummarra, Melita; Baker, Katharine; Ioannou, Liane; Gwini, Stella; Gibson, Stephen; Arnold, Carolyn; Ponsford, Jennie; Cameron, Peter

VERSION 1 – REVIEW

REVIEWER	Catharina Gustavsson, PhD RPT Post doc researcher at Center for Clinical Research Dalarna, Uppsala University, Sweden
REVIEW RETURNED	05-Jun-2017

GENERAL COMMENTS	This paper presents an investigation of an important subject, namely differences in psychological factors, pain and disability between compensable and non-compensable injury, and also association between these psychological factors and worse disability and poorer health. The research question is important and the paper is well worth of publishing. The Introduction section provide a good background for the study. However, I have two mayor concerns about the text that I would like to address. I think the study objectives are imprecisely expressed and a bit confusing. I believe they need to be rephrased to properly explain what was done and what is actually possible to examine by a largely cross-sectional design. I have outlined my comments on the objective in more specific details in the following paragraphs. Also, the study cover a range of variables/factors analysed in several ways and some of the variable being both independent, predictor and/or mediator in different analyses. In total, the study covers an overwhelmingly amount of analyses and tables. It is a bit confusing and difficult to grab and understand for the reader. I suggest the authors could improve the manuscript by rephrasing things in some parts and clarifying aspects in some parts, and thereby making the paper more comprehensible. (Are all tables necessary? Could some tables be removed and only explained in the text? Perhaps displaying the analyses by flow-chars in figures could be one way to make it easier to understand for the reader?) I have a major concern in regard to how the study objectives are expressed in the text. First, the phrasing in the abstract and the main text differs. Please rephrase so that they correspond. Secondly, it is stated that the objectives are to examine “development”, which implies a longitudinal design exploring changes over time. From what I understand the text, the study is largely cross-sectional and
---

thus to examine “associations” and “differences” between variables/factors and groups (compensable vs non-compensable injury) seems appropriate. To further explain my point of view:

- the first part of the study objective, as stated in the Introduction section, i.e. to examine the development of pain, catastrophizing, kinesiophobia and self-efficacy after compensable and non-compensable injury. However, when reading the manuscript I understand that data collection of these variables is done only at 12 months after injury. Is that correct? If no baseline measurement is done of these variables it seems impossible to make conclusions about changes, which is what development supposes. As I understand the text, it would be more proper to say that the study examined differences in these variables between compensable and non-compensable injury at 12 months after injury? I would like that the authors commented on that. Have I got it right or have I misunderstood the text?
- the second part of the objective, as stated in the Introduction section, to my interpretation, concerns the association between these psychological factors and worse disability and poorer health. Both things measured at 12 months post-injury. Also here, the text implies that the objective was to examine development. I consider the study cross-sectional, thus to examine development is impossible.

I am not that very familiar with mediation analyses but I thought that such analyses required a longitudinal design, i.e. at least two time-points of measurement? From what I understand the study is largely cross-sectional with measurements at 12 months post-injury. Most factors/variables (apart from some participant characteristics and injury characteristics including fault attribution) were measured at the same time-point, i.e. at 12 months post-injury, how could you tell what factors preceded another? What mediated what? But, maybe my confusion in this matter is due to insufficient competence in mediation analyses. I suggest some reviewer with adequate competence in mediation analyses should examine this part of the manuscript.

In addition to the overall comments mentioned above, I have some specific comments to the text:

P.2, line 20 (Abstract): The sentence: “Of 732 participants who were referred to the study 82 could not be contacted, and 433 participated and...” leaves the reader wondering about the remaining 217. Please, rephrase. Also, the number 82 is different from the number 70 reported in the results section (P.9, line 36). Which number is correct?

P.2, line 33 (Abstract): I would prefer to name the types of analyses (ordinal, linear and logistic regression) in the methods subtitle, rather than in the results subtitle. Maybe that is my personal preferences, and optional. But in any case, please add information on that also mediation analyses were done.

P.3, line 6-17: the three first bullet points entail somewhat the same information as in the abstract and are not really considerations of strengths and limitations. Please, exclude and possibly include other more relevant statements.

P.5, line 56 – P.6, line 3: The authors write: “The present study administered additional questionnaires about pain and mental health 12-months after injury.” From what I understand the rest of the text, the psychological factors (catastrophizing, kinesiophobia, self-efficacy, injustice) were also assessed by the questionnaire at 12 months? In that case, please include that information in the sentence.

P.7, line 52: The authors write: "The mediating effects of psychological outcomes related to pain were assessed using four measures: the... etcetera". Where these variables really outcomes? How do you know that the participants scores on the measures at 12 months differed from how they would have scored at baseline, and if so, conditional to the injury?

P. 8, line 44: I am not agreeing with the authors that pain interference by BPI and EQ-5D are continuous variables, they are ordinal data. But I am fine with applying a pragmatic approach, and treating variables in the same way as continuous data, if needed for to be able to undertake certain statistical analysis. In such cases I find it appropriate to explain such circumstances. What is the authors' opinion on that?

P.8, line 51-53: The part of the sentence "...linear regression for continuous outcomes (i.e., pain interference and EQ-5D)..." seems as the same information as in the previous sentences? Or is it referring to it separate analyses?

P.10, line 38: I find the text "...were examined with ordinal regression, adjusting for injury severity, age, sex and education..." a bit redundant. Already said in data analysis section and could be removed.

P.12, line 14-16 (Table 2): To my interpretation, the authors report mean and median: "M(Median)" for Pain catastrophizing (PCS) while reporting mean and standard deviation: "M(SD)" for the other variables. I suppose that is due to using Mann-Whitney U for significance testing. However, I suggest that is improper manner of reporting. I suggest that the authors instead should report median and interquartile range: "Md(IQR)" for PCS.

P.18, line 26-28: The authors state that: "Nonetheless, we show for the first time that compensable patients are more likely to also develop lower self-efficacy and higher perceptions of injustice...". I suggest the authors cannot know if the participants developed lower self-efficacy after injury or if they had low self-efficacy already before the injury. From what I understand, self-efficacy or perceived injustice was not measured at early phase after injury (nor before)? If so, a change over time could not be captured. I suggest the authors should temper the statement and say that the compensable patients had lower self-efficacy and higher perceptions of injustice at 12 months as compared to non-compensable.

P.20, line 5-8: The authors write: "Given that self-efficacy and perceived injustice were important predictors and mediators of the relationships between injury and functional outcome...". That is an interesting finding providing important clinical and research implications. But, is it correct that the variables could be both predictors and mediators at the same time? (Question owing to my very limited competence in mediation analyses. As previously said, I suggest some reviewer with adequate competence in mediation analyses should examine the parts of the manuscript that concerns the mediation analyses.)

P.21, line 21-23: at the last page of the manuscript, in the limitations section, the authors address the limitation related to the lack of ability to assume causality due to the largely cross-sectional design. Yes, I agree. But the preceding text does not harmonize with that statement. I suggest the authors revise the presentation of results and conclusions.

REVIEWER	Anabela Silva University of Aveiro, Portugal
REVIEW RETURNED	09-Jun-2017

GENERAL COMMENTS	Congratulations to the authors on this very interesting and very well written paper. Nevertheless, I have a few comments/concerns for clarification. Abstract In the aim, the word “development” leaves the idea that this is a true prospective study and that authors could be sure that variables being assessed were not at all present at baseline. I am not truly convinced of this. I think the aim should be reframed more in line with the more cross-sectional nature of the study; the same applies to the introduction. Design – As the authors say in the limitations section, the study design is “largely cross-sectional” and not prospective; the same applies to the introduction. Methods Page 6 – Why were inclusion criteria different for the two recruitment centers (VSTR and VOTOR)? Was there a difference at baseline between patients from these 2 centers? When was written informed consent given by participants (at 12 months after injury)? Page 7 – line 11: How were body regions defined and counted? Could you present reliability data for the GOS-E and EQ-5D? You used UK norms to interpret EQ-5D scores? Why? Could this have introduced bias somehow? Should it be a limitation? Page 8 – data analysis Line 38 – indicate which were the 4 covariates. Line 40 – explain the sample size better. Include the statistics that are at the end of table 2 Page 9 How did you reach 433 patients: initially there were 732, of which 662 were contacted and 12+97 could not participate. What happen to the remaining patients? Would you please included a flowchart of participants through the study? Table 2 – catastrophizing levels are low? Could this have influenced results? How does it compare with other studies? Figure 1 – Could you add a vertical line crossing 0 (as in Figure2) to help read the table. Discussion End of 1st paragraph: “...compensable patients are more likely to also develop lower self-efficacy...” : “to show” instead of “developing” as it was not measured at baseline one cannot be sure that there were no between group differences at baseline.
---

REVIEWER	Igor Burstyn Drexel University, USA
REVIEW RETURNED	28-Jun-2017

GENERAL COMMENTS	Overarching concerns: Can the paradox be explained by (a) likelihood that compensated injuries are more severe than those that are not compensated and (b) reporting bias/perverse incentive: people who complain of pain are more likely to be compensated and seek compensation, even though the physiological state experienced by those who do and do not seek compensation is the same. The last point is particularly important because it is widely recognized that most people entitled to compensation (by virtue of having work-related injury or disease) never seek compensation; it is the largest source of bias in all this, surely... From this perspective, it is not clear that the study is of "compensable" rather than "compensated" injury, since proportion of compensable that is compensated is widely believed to be tiny. This has to be clarified and discussed in greater detail in the introductory comments that frame presentation of this work. 7-15: I almost missed how the authors determined whether injury was compensable or compensated. This matter must be explained in great detail due to its central role in the paper. The process for determining whether injury was compensable and whether it was compensated, must be described, because most readers do not know how Australian society treats these matters: every country is different with respect to compensating workplace injuries, so describing context of your research is of paramount importance. 21-52-53: The paper concludes with statement about what "should" be done. Such recommendations are always misplaced in reports of observational studies that neither evaluate interventions nor analyze policy implications of such "should's". Please delete. Statistical matters 8-22: List-wise exclusion is a poor practice, unless proportion/number of missing/excluded is very small. At the very least, exploration of mechanism of missingness is needed. In any case, it is better to impute missing values, even mean imputation is better than list-wise deletion. Analysis of pattern of missingness is needed, with particular attention to its evaluation of its dependence on outcomes and mediators. 9-16: Criteria for inclusion of mediators based on "significance" is not explained. What do you mean by "significant"? Is this a reference to some random cut-off of p-value? It is better to focus method on change in effect estimates (point estimates and their SE) when mediator is included or excluded from regressions relating independent and outcome variables. A more clear plan is needed here: evaluate whether independent variables predict mediators, assess whether mediators are related to outcomes, then and only then examine whether mediators alter association of primary interest (independent on outcome). I also recommend including some analyses stratified by values of mediators, as one would in analysis of effect modification in epidemiology. Table 5: Why are only p-values given for direct effects? This makes
--

	it impossible to tell whether purported mediators make tangible difference in effect estimates. This over-reliance on hypothesis tests is misplaced and the authors must show effect estimates and 95%CI for all effects they estimated. P-values should be suppressed because they are not helpful to say the least; please refer to statement by ASA on this matter for full discussion: https://www.amstat.org/asa/files/pdfs/P-ValueStatement.pdf. For example, Suppl Table 3f tells me that indirect and direct effects are not very different in magnitude, e.g. on average -0.1 for fault attribution, despite difference in p-values. Furthermore, claim that disability is “completely mediated” by “perceived injustice” seems to be based on change in p-value from <0.001 to 0.2 (Table 5): this is an inadequate argument and leads to over-interpretation. Discussion is full of such claims and leads me to recommend, in the strongest possible terms, to remove claims of “full mediation” from the discussion because the paper simply is not based on data (and its analysis) that can support such absolutist claims. Such subtleties of the results are best brought to light by not using “significance” as criteria. Given low power of Sobel’s test, modest sample size of author’s study, and problems with summarizing results of any observational study by its p-value, I recommend simply showing effect estimates of independent variables on outcomes with and without inclusion of mediators. Such presentation would give reader the information they need to evaluate what the results mean to them and, consequently, whether they agree with author’s interpretations. Such effect estimates would also help future synthesis of evidence in this arena, beyond trusting qualitative claims of “significance”. These concerns can be FURTHER addressed (not replaced by) by detailed presentation of bootstrapped Sobel-Goodman test. (Why tease the reader with the test and then not give its results?) 21-21-22: It is incorrect to state that causality cannot be inferred from cross-sectional data. This is an often repeated fallacy. Causality can be established when histories are collected in cross-sectional samples. The authors need to focus instead on biases unique to their study, such as differential misclassification and bias that arises from the fact that their subjects KNOW whether their injury was compensated or not, i.e. this is not a study of compensable injuries but that of compensated injuries. Please also note that compensated injuries often produce records that suffer from bias that results from greater interrogation of medical records than injuries that were never subject of adjudication for compensation; all this is detailed in textbooks on occupational epidemiology (e.g. see relevant chapters of Checkoway et al).
--	--

VERSION 1 – AUTHOR RESPONSE

Reviewer 1: Catharina Gustavsson, Post doc researcher at Center for Clinical Research Dalarna, Uppsala University, Sweden.

This paper presents an investigation of an important subject, namely differences in psychological factors, pain and disability between compensable and non-compensable injury, and also association between these psychological factors and worse disability and poorer health. The research question is

important and the paper is well worth of publishing. The Introduction section provide a good background for the study.

RESPONSE: Thank you for your positive comments about the importance of this topic, and the merits of our manuscript.

However, I have two major concerns about the text that I would like to address.

1. I think the study objectives are imprecisely expressed and a bit confusing. I believe they need to be rephrased to properly explain what was done and what is actually possible to examine by a largely cross-sectional design. I have outlined my comments on the objective in more specific details in the following paragraphs.

2. Also, the study cover a range of variables/factors analysed in several ways and some of the variable being both independent, predictor and/or mediator in different analyses. In total, the study covers an overwhelmingly amount of analyses and tables. It is a bit confusing and difficult to grab and understand for the reader. I suggest the authors could improve the manuscript by rephrasing things in some parts and clarifying aspects in some parts, and thereby making the paper more comprehensible. (Are all tables necessary? Could some tables be removed and only explained in the text? Perhaps displaying the analyses by flow-chars in figures could be one way to make it easier to understand for the reader?)

RESPONSE: We agree with the comments made by Reviewer 1 here (and similar comments made by Reviewer 2 below) that the description of the study, and implications of the findings, needed to be rephrased so that they are in line with the cross-sectional nature of the study. We have added a new figure demonstrating the theoretical design and analytic approach of the study (focussing on the mediation-related analyses), as the analytic approach also raised some confusion for Reviewer 3.

We have not removed many results from the manuscript as we feel that each step was fundamental to building the rationale and models of the mediation analyses, and these insights are interesting and meaningful in their own right. However, we hope that the new figure helps to clarify the structure of the results.

I have a major concern in regard to how the study objectives are expressed in the text. First, the phrasing in the abstract and the main text differs. Please rephrase so that they correspond.

RESPONSE: We apologise that the objectives were significantly abbreviated in the abstract due to word restrictions. We have revised the wording of the objectives in both the abstract and introduction so it is consistent, and have cut words elsewhere in the abstract to retain <300 words.

Secondly, it is stated that the objectives are to examine “development”, which implies a longitudinal design exploring changes over time. From what I understand the text, the study is largely cross-sectional and thus to examine “associations” and “differences” between variables/factors and groups (compensable vs non-compensable injury) seems appropriate. To further explain my point of view:

- The first part of the study objective, as stated in the Introduction section, i.e. to examine the development of pain, catastrophizing, kinesiophobia and self-efficacy after compensable and non-compensable injury. However, when reading the manuscript I understand that data collection of these variables is done only at 12 months after injury. Is that correct? If no baseline measurement is done of these variables it seems impossible to make conclusions about changes, which is what development supposes. As I understand the text, it would be more proper to say that the study examined differences in these variables between compensable and non-compensable injury at 12 months after injury? I would like that the authors commented on that. Have I got it right or have I misunderstood the text?

- The second part of the objective, as stated in the Introduction section, to my interpretation, concerns the association between these psychological factors and worse disability and poorer health. Both things measured at 12 months post-injury. Also here, the text implies that the objective was to examine development. I consider the study cross-sectional, thus to examine development is impossible.

I am not that very familiar with mediation analyses but I thought that such analyses required a longitudinal design, i.e. at least two time-points of measurement? From what I understand the study is largely cross-sectional with measurements at 12 months post-injury. Most factors/variables (apart from some participant characteristics and injury characteristics including fault attribution) were measured at the same time-point, i.e. at 12 months post-injury, how could you tell what factors preceded another? What mediated what? But, maybe my confusion in this matter is due to insufficient competence in mediation analyses. I suggest some reviewer with adequate competence in mediation analyses should examine this part of the manuscript.

RESPONSE: We have removed all terminology implying “development” as we did not measure change over time. We wish to clarify for Reviewer 1 that the use of mediation analysis in a cross-sectional cohort study is acceptable, especially when there is a strong theoretical basis for the sequential or “causal” associations between factors. Of course mediation analysis of cross-sectional data should not be interpreted as a time-sensitive causal relationship over time. In the present study, the independent variables (compensation, fault attribution), mediators (i.e., PSEQ, PCS, IEQ) and dependent variables (BPI interference, Health Status, Disability) were all measured and/or established at one year post-injury. The mediation analysis and results in our observational study therefore provide us with a hypothesis generating model that could be tested for causal/mechanistic effects in future interventions studies. The present models, as with all mediation designs, were built on the theoretical assumption that the mediators are mechanistically associated with the dependent variable, and that the association between the independent variable with the dependent variable arises due to its relationship with (or “influence on”) the mediating variable. Given the cross-sectional design we must, of course, be cautious in interpreting these relationships as not being truly causal as the relationship between mediating variable and dependent variables is often bidirectional (see discussion in Shrout and Bolger, 2002).

We have highlighted this key point on Page 22, and emphasise that future studies are required to examine whether bringing about change in the mediating variable (e.g., through an intervention) leads to a reduction in the magnitude of the direct and/or indirect effects between the independent and dependent variables.

In addition to the overall comments mentioned above, I have some specific comments to the text:

• P.2, line 20 (Abstract): The sentence: “Of 732 participants who were referred to the study 82 could not be contacted, and 433 participated and...” leaves the reader wondering about the remaining 217. Please, rephrase. Also, the number 82 is different from the number 70 reported in the results section (P.9, line 36). Which number is correct?

RESPONSE: We have provided further detail on the number of potential participants that declined to participate in the abstract. Specifically, 70 participants could not be contacted, and 12 did not meet the eligibility criteria (i.e., a total of 82 potential participants who should not be considered in the response rate). We also wish to clarify that we had forgotten to account for 120 participants who gave either verbal or written consent, but did not complete the questionnaires. We have updated these details in the Results on Page 12.

• P.2, line 33 (Abstract): I would prefer to name the types of analyses (ordinal, linear and logistic regression) in the methods subtitle, rather than in the results subtitle. Maybe that is my personal preferences, and optional. But in any case, please add information on that also mediation analyses were done.

RESPONSE: We have made these changes to the abstract.

• P.3, line 6-17: the three first bullet points entail somewhat the same information as in the abstract and are not really considerations of strengths and limitations. Please, exclude and possibly include other more relevant statements.

RESPONSE: The bullet points have been rewritten as suggested.

• P.5, line 56 – P.6, line 3: The authors write: “The present study administered additional questionnaires about pain and mental health 12-months after injury.” From what I understand the rest of the text, the psychological factors (catastrophizing, kinesiophobia, self-efficacy, injustice) were also assessed by the questionnaire at 12 months? In that case, please include that information in the sentence.

RESPONSE: These details have been added to the methods (Page 6).

• P.7, line 52: The authors write: “The mediating effects of psychological outcomes related to pain were assessed using four measures: the... etcetera”. Where these variables really outcomes? How do you know that the participants scores on the measures at 12 months differed from how they would have scored at baseline, and if so, conditional to the injury?

RESPONSE: We have replaced the word “outcomes” with “characteristics” in this sentence, and made consistent revisions elsewhere in the manuscript.

• P. 8, line 44: I am not agreeing with the authors that pain interference by BPI and EQ-5D are continuous variables, they are ordinal data. But I am fine with applying a pragmatic approach, and treating variables in the same way as continuous data, if needed for to be able to undertake certain statistical analysis. In such cases I find it appropriate to explain such circumstances. What is the authors' opinion on that?

RESPONSE: It is true that the individual items on the BPI can be treated as ordinal as they are measured using a numerical rating scale (0 - 10); however, we analysed the interference subscale of the BPI, which is the average rating across seven ratings. The interference subscale can therefore be any number between zero and ten (including decimal points). We understand that this confusion may have come about because we generated ordinal groupings for the BPI severity scores (which are also an average score across four ratings of pain intensity). This was done for the severity subscale because a relatively large proportion of the sample reported no pain at all on the BPI, and the skew in the pain severity data made analysis and interpretation of linear regression more challenging. We therefore examined pain severity as an ordinal outcome (no pain, low pain, moderate-severe pain) using ordinal regression, but then examined pain interference only in patients who experienced some pain.

For the EQ-5D, again, although the individual items relating to the severity of subjective problems with mobility, pain or discomfort, anxiety or depression etc. are rated on ordinal three level scales, the EQ-5D summary score that we analysed generates an overall indication of health status ranging from 0.00 to 1.00 (i.e., this is a continuous variable, not ordinal).

• P.8, line 51-53: The part of the sentence “...linear regression for continuous outcomes (i.e., pain interference and EQ-5D)...” seems as the same information as in the previous sentences? Or is it referring to it separate analyses?

RESPONSE: This is correct. We have deleted the first sentence, as this information is summarised more succinctly in the sentence beginning with “Ordinal regression was used...”

• P.10, line 38: I find the text “...were examined with ordinal regression, adjusting for injury severity, age, sex and education...” a bit redundant. Already said in data analysis section and could be removed.

RESPONSE: The first sentence of this paragraph has been truncated to read “Relationships between baseline injury characteristics and pain severity one year post-injury are reported in Table 3.”

• P.12, line 14-16 (Table 2): To my interpretation, the authors report mean and median: “M(Median)” for Pain catastrophizing (PCS) while reporting mean and standard deviation: “M(SD)” for the other variables. I suppose that is due to using Mann-Whitney U for significance testing. However, I suggest that is improper manner of reporting. I suggest that the authors instead should report median and interquartile range: “Md(IQR)” for PCS.

RESPONSE: We have made the suggested changes to Table 2. We also provide clearer definitions of the abbreviations in the table, and the data presented, in the Table footnotes.

• P.18, line 26-28: The authors state that: “Nonetheless, we show for the first time that compensable patients are more likely to also develop lower self-efficacy and higher perceptions of injustice...”. I suggest the authors cannot know if the participants developed lower self-efficacy after injury or if they had low self-efficacy already before the injury. From what I understand, self-efficacy or perceived injustice was not measured at early phase after injury (nor before)? If so, a change over time could not be captured. I suggest the authors should temper the statement and say that the compensable patients had lower self-efficacy and higher perceptions of injustice at 12 months as compared to non-compensable.

RESPONSE: We agree that the use of “develop” type language is inappropriate for this particular study. We have revised the sentence to read as follows:

“...we have shown for the first time that lower self-efficacy and higher perceptions of injustice after compensable injury mediated the degree to which pain impacted on a range of daily activities, as well as health and disability outcomes at one year after injury.”

• P.20, line 5-8: The authors write: “Given that self-efficacy and perceived injustice were important predictors and mediators of the relationships between injury and functional outcome...”. That is an interesting finding providing important clinical and research implications. But, is it correct that the variables could be both predictors and mediators at the same time? (Question owing to my very limited competence in mediation analyses. As previously said, I suggest some reviewer with adequate competence in mediation analyses should examine the parts of the manuscript that concerns the mediation analyses.)

RESPONSE: One can indeed analyse the same variable as a “predictor” (or independent variable) and as a “mediator”. In fact, according to the criteria for mediation analysis, these preliminary steps are necessary to identify whether mediation analysis should proceed.

That is, for mediation, there must be an association between the independent variable and the mediator, and between the mediator and the dependent variable (clarification has been added to the methods, Page 11, with the sentence “The strength of indirect effects was only examined if the proposed mediator was significantly associated with both the independent and dependent variables in preliminary linear and logistic regression analyses”).

In the present study, the linear/ordinal/logistic regressions simply examined the direct relationships for path a (between IV and MV) and for path b (between MV and DV). We have revised the subheadings in the results section so that it is clear where we are talking about the testing of these a priori

assumptions for the mediation analysis, and also emphasise these sequential considerations in Figure 1.

To avoid confusion around the causal inferences in this study, we have removed the term “predictor” from the manuscript, and replaced it with “independent variable”.

• P.21, line 21-23: at the last page of the manuscript, in the limitations section, the authors address the limitation related to the lack of ability to assume causality due to the largely cross-sectional design. Yes, I agree. But the preceding text does not harmonize with that statement. I suggest the authors revise the presentation of results and conclusions.

RESPONSE: Reviewer 3 also commented on our statements on assuming causality from cross-sectional studies. While they say that such causal associations can often still be made in cross-sectional studies, they recommended that we focus more on the biases specific to our study (e.g., differential misclassification and bias when subjects know if their injury was compensated or not). We have created a new paragraph break at this point of the discussion, and focus on the fact that the “outcomes” were measured at the same time, and therefore causality between the mediator and dependent variables cannot be assumed.

Reviewer 2: Anabela Silva, University of Aveiro, Portugal

Congratulations to the authors on this very interesting and very well written paper. Nevertheless, I have a few comments/concerns for clarification.

RESPONSE: Thank you for your positive comments about the importance and quality of our paper.

Abstract

In the aim, the word “development” leaves the idea that this is a true prospective study and that authors could be sure that variables being assessed were not at all present at baseline. I am not truly convinced of this. I think the aim should be reframed more in line with the more cross-sectional nature of the study; the same applies to the introduction.

RESPONSE: We agree with this concern, and have amended the whole manuscript, including the abstract, to avoid any inferences that the outcomes may have been measured over multiple time points.

Design

- As the authors say in the limitations section, the study design is “largely cross-sectional” and not prospective; the same applies to the introduction.

RESPONSE: The description of the study design has been revised in the Abstract (Page 2) and Introduction (Page 5) as an “observational registry-based cohort study”.

Methods and results

- Page 6 – Why were inclusion criteria different for the two recruitment centers (VSTR and VOTOR)? Was there a difference at baseline between patients from these 2 centers?

RESPONSE: We wish to clarify a few important points here, which we have also clarified when describing the registries on Pages 6 to 7 of the manuscript.

- There was one “centre” in which the study was conducted. All patients had been admitted to a single major trauma hospital in Victoria.

- The patients were identified through two registry sources (not “centres”): The Victorian State Trauma Registry (VSTR), and the Victorian Orthopaedic Trauma Outcomes Registry (VOTOR).

- Both registries are managed in the Department of Epidemiology and Preventive Medicine, Monash University, in the Pre-hospital, emergency and Trauma research group (where authors Melita Giummarra and Peter Cameron are based). The same interviewing staff work on both VSTR and VOTOR, and the same follow-up interviews are conducted for each registry.

- While the study “centre” was a single location (i.e., The Alfred Hospital, Melbourne, Victoria), the nature and focus of each registry differs, and this is why the inclusion criteria differ when we describe VSTR and VOTOR in the manuscript. In brief:

- o VSTR only comprises patients who meet major trauma criteria, and is a population based registry that represents all major trauma cases that resulted in admission to a trauma service (i.e., at 138 hospitals) in the state of Victoria.

- o VOTOR comprises patients who had orthopaedic injuries that may, or may not, have met major trauma criteria. Given the markedly higher incidence of less severe orthopaedic injuries, VOTOR is a “sentinel” registry and only includes patients admitted to four hospitals in Victoria. Patients can be included in both the VOTOR and VSTR registry if they meet the inclusion criteria for both registries.

Further details on the number of patients recruited from the two registries, and any differences in injury severity, or study outcomes, between participants were recruited from VSTR (some of whom were also on VOTOR) versus patients who were recruited from VOTOR only is provided on Page 12-11.

- When was written informed consent given by participants (at 12 months after injury)?

RESPONSE: The following detail about consent for the registries is now provided on Page 6-7:

“Patients are provided with information about the registries before the first follow-up interview, and are given the opportunity to opt-off. Less than one percent of patients elect to be removed from VOTOR or VSTR.”

We explain that participants were invited to participate in the 12-month registry interview (Page 7, under “Materials and Procedures”), and that participants gave written informed consent for this specific study after that invitation.

- Page 7 – line 11: How were body regions defined and counted?

RESPONSE: As per the scoring guidelines for the Abbreviated Injury Scale, severity scores are given for the specific affected body region in order to facilitate calculation of the Injury Severity Score (i.e., the sum of the most severe AIS scores squared, from three different body regions). Using this system, injuries are classified into the regions of: head; face; neck; thorax; abdomen; spine; upper extremity; lower extremity; unspecified. As the AIS/ISS scoring systems may not be familiar to all BMJ Open readers, we have provided additional information about the AIS scoring method, and body regions in the Methods (Page 8).

- Could you present reliability data for the GOS-E and EQ-5D?

RESPONSE: Details on the reliability and validity of the GOS-E and EQ-5D have been added on Page 9.

- You used UK norms to interpret EQ-5D scores? Why? Could this have introduced bias somehow? Should it be a limitation?

RESPONSE: The UK values/tarrifs for the EQ-5D were used as there are no validated Australian values (although our group is currently working towards establishing Australian-based values). A systematic review of outcome studies using the EQ-5D found that the UK tarrifs were most commonly used, and are considered appropriate in Australian studies given that Australia and the UIK have relatively similar cultural, economic and health system characteristics. We have explained this further in the Methods on Page 9. We do not feel that this is a limitation, and is not typically noted as a limitation in other studies from the VSTR or VOTOR (e.g., a recent paper by Gabbe et al published in BMJ Open).

Data analysis

- Line 38 – indicate the 4 covariates.

RESPONSE: The covariates were age, sex, pain severity, and injury severity (number of body regions with a moderate-severe AIS score). These details have been added on Page 10), and have also been clarified in each of the Table and Figure footnotes.

- Line 40 – explain the sample size better.

RESPONSE: We have moved the sample size description to the end of the Data Analytic Approach section of the Methods, and revised the explanation of the sample size requirements for the respective analyses conducted.

- Include the statistics that are at the end of table 2

RESPONSE: In line with comments from Reviewer 1 about the types of statistics reported in Table 2, we have updated the data for the PCS, and provide clearer notes on the statistics reported in Table 2 in the table footnotes.

- Page 9, How did you reach 433 patients: initially there were 732, of which 662 were contacted and 12+97 could not participate. What happen to the remaining patients? Would you please included a flowchart of participants through the study?

RESPONSE: Thank you for this question, this was an oversight on our behalf. We had forgotten to account for the 120 potential participants who had given either verbal or written consent, but then did not complete the questionnaires. We have corrected the numbers in text (see Abstract, and Results, Page 12).

- Table 2 – catastrophizing levels are low? Could this have influenced results? How does it compare with other studies?

RESPONSE: We have added some discussed on this point on Page 23-24.

- Figure 1 – Could you add a vertical line crossing 0 (as in Figure 2) to help read the table.

RESPONSE: Vertical lines have been added to Figure 1 (now Figure 2) as suggested.

Discussion:

• End of 1st paragraph: "...compensable patients are more likely to also develop lower self-efficacy..." : "to show" instead of "developing" as it was not measured at baseline one cannot be sure that there were no between group differences at baseline.

RESPONSE: We agree with this point and now state the following:

"...we have shown for the first time that lower self-efficacy and higher perceptions of injustice after compensable injury mediated the degree to which pain impacted on a range of daily activities, as well as health and disability outcomes at one year after injury." (Page 22).

All use of language suggesting "development" has been removed throughout the manuscript.

Reviewer 3: Igor Burstyn, Drexel University, USA.

Overarching concerns:

Can the paradox be explained by (a) likelihood that compensated injuries are more severe than those that are not compensated and (b) reporting bias/perverse incentive: people who complain of pain are more likely to be compensated and seek compensation, even though the physiological state experienced by those who do and do not seek compensation is the same. The last point is particularly important because it is widely recognized that most people entitled to compensation (by virtue of having work-related injury or disease) never seek compensation; it is the largest source of bias in all this, surely... From this perspective, it is not clear that the study is of "compensable" rather than "compensated" injury, since proportion of compensable that is compensated is widely believed to be tiny. This has to be clarified and discussed in greater detail in the introductory comments that frame presentation of this work.

RESPONSE: These are excellent points, which we respond to in turn below. First, however, we wish to briefly clarify that this study classified participants as "compensable" if they were admitted to hospital with a linked compensation claim number and/or indicated at the 12-month interview that their injury was compensable (i.e., the injury mechanism and cause met the eligibility criteria for a work or transport-injury compensation claim). The study therefore applied classifications of "compensable" rather than "compensated" status. We have provided a brief definition of compensable injury in the first paragraph of the Introduction (Page 4), and in the Methods (Page 8).

Before responding to the overarching comments above, we also wish to clarify upfront that having a compensation claim in the present setting (Victoria, Australia) only results in lump sum "damages" or "compensation payments" for a minority of cases. For most injured persons in this setting a compensation claim fundamentally provides income support and/or covers all or part of the cost of medical and rehabilitation care, and long-term costs supporting independence (e.g., home modifications, carer costs etc.). We have clarified this in paragraph one of the Introduction (Page 4) as this is a frequent misconception that having a compensation claim constitutes "being compensated" and receiving financial benefit.

1. People who make a claim have more severe injuries or worse outcomes

This concern is very valid when applied to the vast majority of research on compensable injury outcomes around the world. This paradox has been partly described by the "reverse causality"

hypothesis (see [http://www.jclinepi.com/article/S0895-4356\(12\)00187-4/abstract](http://www.jclinepi.com/article/S0895-4356(12)00187-4/abstract)). That is, in most jurisdictions those who have poorer outcomes are more likely to lodge a compensation claim, or seek compensation via litigation. Especially where only a subset of cases are eligible for compensation (e.g., if another was at fault).

While this may apply to a minority of cases in the present setting (Victoria, Australia), the vast majority of those who are hospitalized following injury will have a compensation claim if they are eligible, especially after a transport injury. This is facilitated by the fact that the regionalized trauma system in Victoria has established systematic collaboration between the ambulance and retrieval services, trauma centres, and the compensation system (Transport Accident Commission). Patients who require transfer to and/or hospital admission typically automatically meet the medical excess threshold requirements for a compensation claim and meet the eligibility requirements for a claim (defined below), then lodgment of that claim is facilitated through the patient admission procedures.

In Victoria, all persons are eligible for compensation, regardless of their role in the injury (i.e., “no fault”), if they are injured in an incident involving a motorized vehicle that could be registered (but may not be registered at the time; i.e., excluding home constructed/designed motorized vehicles), or a vehicle that operates on rails (i.e., trains/trams). Likewise, all Victorian employees are entitled to lodge a compensation claim regardless of who was at fault for their injury. Therefore the only category of work or transport injury trauma patients who are typically ineligible are those who were injured in a bicycle crash if there was no involvement of a motorized/rail vehicle.

All of this said, we do know through our routine linkage of the trauma registry with the compensation system that a handful of cases who were eligible for compensation do not proceed with a claim after hospitalization for transport-related injury. These tend to be motorcyclists, who we (and the TAC) assume may not wish to be identified and potentially tracked by a regulated government health/welfare integrated system.

We have introduced the reverse causality explanation in the Introduction (Page 4), and emphasize the key system design features as a strength of the study setting in the discussion (Page 25).

2. People complaining of worse pain (or other problems) after injury are more likely to be compensated and to seek compensation.

This point ties into the literature focused on the potential exaggeration of outcomes in order to receive “secondary gain” after injury. We acknowledge that this no doubt happens in all contexts after compensable injury, especially if it becomes possible for one to receive income support/disability pension. Some studies have estimated that this may occur to varying degrees in around 30% of cases (Bass C, Halligan P. Factitious disorders and malingering: challenges for clinical assessment and management. *Lancet*.2014;383(9926):1422-1432). That said, it should be noted that the present data were not collected from within the compensation system, participants were aware that their individual data would not be shared with the compensation system (or any other potentially interested parties), and participants were not explicitly informed of the study hypotheses or research questions. Rather, all of outcomes data were collected independently of the trauma and compensation system by interviewers employed by the Victorian State Trauma Outcomes Monitoring Group, and by the specific study staff. It is unlikely, therefore, that participants would augment their responses in line with concerns about seeking secondary gain. We have explicitly discussed and expanded on the potential role of these factors in the Introduction (Page 4), and note these key points when describing the study procedures (Page 7).

• 7-15: I almost missed how the authors determined whether injury was compensable or compensated. This matter must be explained in great detail due to its central role in the paper. The process for determining whether injury was compensable and whether it was compensated, must be described, because most readers do not know how Australian society treats these matters: every country is different with respect to compensating workplace injuries, so describing context of your research is of paramount importance.

RESPONSE: The study defined injuries as compensable, not compensated. We have linked the present compensable cases to the compensation payments data from the respective compensation systems in Victoria (Transport Accident Commission; WorkSafe Victoria), and only a small number of cases did not end up having a compensation claim. We have not included these data in the present manuscript as they will be analysed and discussed in another paper currently under preparation. Instead, a full definition of “compensable injury” as it was applied for these analyses has been added to the methods (Page 8).

• 21-52-53: The paper concludes with statement about what “should” be done. Such recommendations are always misplaced in reports of observational studies that neither evaluate interventions nor analyze policy implications of such “should’s”. Please delete.

RESPONSE: The concluding statement has been deleted as suggested.

Statistical matters

• 8-22: List-wise exclusion is a poor practice, unless proportion/number of missing/excluded is very small. At the very least, exploration of mechanism of missingness is needed. In any case, it is better to impute missing values, even mean imputation is better than list-wise deletion. Analysis of pattern of missingness is needed, with particular attention to its evaluation of its dependence on outcomes and mediators.

RESPONSE: This statement was included in the methods for full transparency of approaches, but we wish to reassure the reviewer that there was very little missing data in the present study (<5% of cases). This has been clarified on Page 10.

• 9-16: Criteria for inclusion of mediators based no “significance” is not explained. What do you mean by “significant”? Is this a reference to some random cut-off of p-value? It is better to focus method on change in effect estimates (point estimates and their SE) when mediator is included or excluded from regressions relating independent and outcome variables.

A more clear plain is needed here: evaluate whether independent variables predict mediators, assess whether mediators are related to outcomes, then and only then examine whether mediators alter association of primary interest (independent on outcome).

RESPONSE: The criteria for significance are outlined at the beginning of the Data analytic approach section of the methods, and we have added further detail as follows:

“Significance was determined if $\alpha < 0.05$, or if the 95% confidence did not include 1.00 (logistic and ordinal regression) or 0.00 (linear regression, mediation).”

We did follow the criteria for mediation suggested, and have explained these assumptions and procedures more clearly on Page 11, as follows:

“The strength of indirect effects was only examined if the proposed mediator was significantly associated with both the independent and dependent variables in preliminary linear and logistic regression analyses (Shrout & Bolger, 2002)”.

“The presence and strength of indirect (i.e., mediated) effects were determined from examination of the size of the coefficient, and the bootstrapped 95% confidence intervals such that effects were considered significant if the CI did not contain zero... Effect estimates were interpreted as very small (<.01), small (>.20), moderate (>.50), large (>.80), very large (>1.20) or huge (>2.0).”

We have updated Table 5 so that it reports the effects and 95% CIs for the indirect and direct effects (accounting for the indirect/mediated path) of each model.

- I also recommend including some analyses stratified by values of mediators, as one would in analysis of effect modification in epidemiology.

RESPONSE: We believe that these analyses would be appropriate if our study had taken a moderation design (i.e., examining when and how a third variable influences the direction or strength of the relationship between an IV and a DV). However, we were interested in whether the mediator variables accounted for the relationship between the IV and DV, requiring a mediation design. These conceptual factors have been described in Baron and Kenny (1986), and several other texts/papers on moderation versus mediation. In this instance, we have therefore chosen not to include further analysis of effect modification in the revised manuscript.

- Table 5: Why are only p-values given for direct effects? This makes it impossible to tell whether purported mediators make tangible difference in effect estimates. This over-reliance on hypothesis tests is misplaced and the authors must show effect estimates and 95%CI for all effects they estimated. P-values should be suppressed because they are not helpful to say the least; please refer to statement by ASA on this matter for full discussion: <https://www.amstat.org/asa/files/pdfs/P-ValueStatement.pdf>. For example, Suppl Table 3f tells me that indirect and direct effects are not very different in magnitude, e.g. on average -0.1 for fault attribution, despite difference in p-values.

RESPONSE: We have removed the p-values from Table 5, and report the effects and 95% CIs for the indirect and direct effects (accounting for the indirect/mediated path) of each model instead.

- Furthermore, claim that disability is “completely mediated” by “perceived injustice” seems to be based on change in p-value from <0.001 to 0.2 (Table 5): this is an inadequate argument and leads to over-interpretation.

- Discussion is full of such claims and leads me to recommend, in the strongest possible terms, to remove claims of “full mediation” from the discussion because the paper simply is not based on data (and its analysis) that can support such absolutist claims. Such subtleties of the results are best brought to light by not using “significance” as criteria.

RESPONSE: The language that we had used to describe the type of mediated relationship is consistent with published methods papers on conducting and interpreting mediation (e.g., Shrout and Bolger, 2002). We have modified our phrasing slightly (e.g., from terms like “completely mediated” or “fully mediated” to “complete mediation” so that we are 100% consistent with the terms used previously by Shrout and Bolger. We have included a definition of these terms in the Methods to reduce the likelihood that readers will misinterpret the results (Page 13). In the discussion we have also revised any relevant statements with clearer lay descriptions of the findings, such as:

“This study demonstrated an association between compensable injury and worse disability and health outcomes (i.e. in relation to mobility, self-care, activity participation, pain, and anxiety/depression), which were to varying degrees attributable to the experience of lower self-efficacy and higher injustice perceptions.” (in place of “... which was fully mediated by self-efficacy”).

Given low power of Sobel’s test, modest sample size of author’s study, and problems with summarizing results of any observational study by its p-value, I recommend simply showing effect estimates of independent variables on outcomes with and without inclusion of mediators. Such presentation would give reader the information they need to evaluate what the results mean to them and, consequently, whether they agree with author’s interpretations. Such effect estimates would also help future synthesis of evidence in this arena, beyond trusting qualitative claims of “significance”. These concerns can be FURTHER addressed (not replaced) by detailed presentation of bootstrapped Sobel-Goodman test. (Why tease the reader with the test and then not give its results?)

RESPONSE: We now focus the reporting of indirect effects on the size of the coefficients of both the indirect effect, and strength and significance (based on the 95% CI) of the direct effect. Instructions on how to interpret the 95% CI have been added to the table and figure footnotes as many readers rely on p-values to understand whether an effect is significant. We also wish to remind the reviewer that these 95% CIs are based on 500 resamples, thereby significantly improving the power to detect indirect effects (see Shrout and Bolger, 2002).

21-21-22: It is incorrect to state that causality cannot be inferred from cross-sectional data. This is an often repeated fallacy. Causality can be established when histories are collected in cross-sectional samples. The authors need to focus instead on biases unique to their study, such as differential misclassification and bias that arises from the fact that their subjects KNOW whether their injury was compensated or not, i.e. this is not a study of compensable injuries but that of compensated injuries. Please also note that compensated injuries often produce records that suffer from bias that results from greater interrogation of medical records than injuries that were never subject of adjudication for compensation; all this is detailed in textbooks on occupational epidemiology (e.g. see relevant chapters of Checkoway et al).

RESPONSE: As per our explanation above, we acknowledge the problems relating to causality/bias in the compensation literature; however, we believe that this specific point is not so pertinent given the present recruitment strategy (i.e., hospitalised injuries), and the way that the trauma system integrates

with the State compensation systems in Victoria, Australia. In the present study the bigger problem is the fact that hospitalised compensable injuries tend to be more severe than non-compensable injuries (largely because they are caused by transport injury). To attenuate the impact of injury severity on the findings we included injury severity as a covariate in all analyses.

VERSION 2 – REVIEW

REVIEWER	Catharina Gustavsson, PhD RPT Center for Clinical Research Dalarna, Uppsala University, Sweden.
REVIEW RETURNED	08-Aug-2017

GENERAL COMMENTS	I have read the resubmitted version of the manuscript and the authors responses to the comments made by me and the other reviewers. I consider that the authors have satisfactorily responded to the comments and made adequate revisions to the manuscript. I regard the manuscript acceptable for publication.
--

REVIEWER	Anabela G. Silva School of Health Sciences University of Aveiro
REVIEW RETURNED	14-Aug-2017

GENERAL COMMENTS	The authors have addressed my previous queries and I have no further questions.
---